# Pax6 organizes the anterior eye segment by guiding two distinct neural crest waves

Masanari Takamiya[1], Johannes Stegmaier[2¤a], Andrei Yu Kobitski[1,3], Benjamin Schott[2], Benjamin D. Weger[1], Dimitra Margariti[1], Angel R. Cereceda Delgado[1], Victor Gourain[1], Tim Scherr[2], Lixin Yang[1], Sebastian Sorge[1¤b], Jens C. Otte[1], Volker Hartmann[4], Jos van Wezel[5], Rainer Stotzka[4], Thomas Reinhard[6], Günther Schlunck[6], Thomas Dickmeis[1], Sepand Rastegar[1], Ralf Mikut[2], Gerd Ulrich Nienhaus[1,3,7,8], Uwe Strähle[1] *

1 Institute of Biological and Chemical Systems - Biological Information Processing, Karlsruhe Institute of Technology, Karlsruhe, Germany, 2 Institute for Automation and Applied Informatics, Karlsruhe Institute of Technology, Karlsruhe, Germany, 3 Institute of Applied Physics, Karlsruhe Institute of Technology, Karlsruhe, Germany, 4 Institute for Data Processing and Electronics, Karlsruhe Institute of Technology, Karlsruhe, Germany, 5 Steinbuch Centre for Computing, Karlsruhe Institute of Technology, Karlsruhe, Germany, 6 Eye Center, Freiburg University Medical Center, Freiburg, Germany, 7 Institute of Nanotechnology, Karlsruhe Institute of Technology, Karlsruhe, Germany, 8 Department of Physics, University of Illinois at Urbana-Champaign, Urbana, Illinois, United States of America

¤a Current address: Institute of Imaging & Computer Vision, RWTH Aachen University, Aachen, Germany
¤b Current address: The Francis Crick Institute, London, United Kingdom
* uwe.straehle@kit.edu

**Data Availability Statement:** All relevant data are within the manuscript and its Supporting Information files.

## Abstract

Cranial neural crest (NC) contributes to the developing vertebrate eye. By multidimensional, quantitative imaging, we traced the origin of the ocular NC cells to two distinct NC populations that differ in the maintenance of *sox10* expression, Wnt signalling, origin, route, mode and destination of migration. The first NC population migrates to the proximal and the second NC cell group populates the distal (anterior) part of the eye. By analysing zebrafish *pax6a/b* compound mutants presenting anterior segment dysgenesis, we demonstrate that Pax6a/b guide the two NC populations to distinct proximodistal locations. We further provide evidence that the lens whose formation is *pax6a/b*-dependent and lens-derived TGFβ signals contribute to the building of the anterior segment. Taken together, our results reveal multiple roles of Pax6a/b in the control of NC cells during development of the anterior segment.

## Author summary

Neural crest (NC) cells are pluripotent embryonic stem cells that originate from neural tissue during early embryogenesis. The origin and fate of NC cells have been studied intensively in various animal species showing that the eye receives input from NC cells. There are several inherited diseases like aniridia, Peters anomaly, Axenfeld-Rieger syndrome and congenital glaucoma affecting structures of the eye derived from NC cells. As of yet, it has remained unclear how the congenital abnormalities of the eye develop in these

**Funding:** This work was supported by EU FP7-HEALTH-2007-B2 NeuroXsys (https://cordis.europa.eu/project/rcn/91045/factsheet/en; U.S.), the Interreg NSB-Upper Rhine (http://www.interreg-upperrhine.eu/; U.S.), the Helmholtz-Gemeinschaft Programs BioInterfaces (https://www.helmholtz.de/forschung/schluesseltechnologien/biointerfaces_in_technology_and_medicine/; U.S.) and Science and Technology of Nanosystems (https://www.helmholtz.de/forschung/schluesseltechnologien/science_and_technology_of_nanosystems/; U.S. and G.U.N.) at KIT, EU IP ZF-HEALTH (https://cordis.europa.eu/project/rcn/95260/reporting/en; U.S.), BMBF-Molecular Interaction Engineering (https://www.forschung-mie.de/; U.S.), DFG SPP 1736 Algorithms for Big Data (https://www.dfg.de/en/funded_projects/current_projects_programmes/list/projectdetails/index.jsp?id=237179235&sort=nr_asc&prg=SPP; J.S.), DFG Graduiertenkolleg 2039 (https://www.dfg.de/gefoerderte_projekte/programme_und_projekte/listen/projektdetails/index.jsp?id=250526013; U.S. and G.U.N.) and Helmholtz Network of Excellence "AESC - Algorithm Engineering for the Scalability Challenge: Algorithms Driving the Information Society" (ExNet-0033-Phase 3; U.S., G.U.N. and R.M.). The funders had no role in study design, data collection and analysis, decision to publish, or preparation of the manuscript.

**Competing interests:** The authors have declared that no competing interests exist.

patients. We have applied genetic tools and multidimensional live imaging to study the dynamics of NC cell behaviour during eye formation. We found that two waves of NC cell populations with different properties contribute to distinct parts of the eye. By using zebrafish mutants as disease models for human anterior segment dysgenesis, we demonstrate that proper guidance of the two NC populations is crucial for a normal development of the anterior segment of the eye.

## Introduction

Neural crest (NC) cells are a population of pluripotent embryonic stem cells that originate at the boundary between neural and non-neural ectoderm. Subsequently, NC cells delaminate from the neuroectoderm and migrate to remote regions of the body to give rise to a wide variety of different cell types such as the peripheral nervous system, the cranial skeleton and ocular tissues [1].

Anterior segment (AS) dysgenesis (ASD) of the eye refers to a group of developmental disorders of the distal structures of the eye such as the cornea, iris, and trabecular meshwork [2]. These ocular structures as well as the sclera are all NC derivatives [1,3]. ASD is associated with 8 human genes, including the transcription factors PITX2, FOXC1/2 and paired box protein Pax-6 (PAX6) [2,4]. Defective TGFβ signalling also causes ASD: TGFβ2 is required for *Pitx2* and *Foxc1* expression in NC-derived AS structures in mice [5,6]. The lens which expresses TGFβ2 was shown to act as a signalling centre that organizes AS morphogenesis. Homozygous mutant studies in rodents indicated a non-cell autonomous role of PAX6 in guiding NC cells into the eye [7] and heterozygous conditional mutation of PAX6 in the lens and the cornea led to ASD with malformed trabecular meshwork and Schlemm's canal [8]. Although the defects are initially evident mainly in the AS, PAX6 mutant mice developed elevated intraocular pressure (IOP) which later led to damage of the whole retina as a result of juvenile glaucoma.

In mammalian eye development, reduced PAX6 levels correlate with the severity of eye defects [9]. Pax6 expression establishes a concentration gradient, with higher levels on the distal side of the optic cup (i.e., the cornea, lens and iris) and lower levels on the proximal side [10,11]. Proximal marker gene expression in Pax6 homozygous mutant eyes was expanded abnormally toward the distal side, with no nasal-temporal and dorsal identities [10,12]. In zebrafish, signals from the midline induce the expression of Pax2, which inhibits Pax6 expression in the optic stalk and thereby regulates the partitioning of the optic primordium [13]. The AS of the zebrafish eye is not only structured similarly to that of mammals [14,15] but also displays the same *pax6* dependency. *pax6b* mutants show a spectrum of ASD phenotypes with misregulated AS marker gene expression [16].

Here, we have examined how neural crest cells contribute to the zebrafish eye, and whether this contribution involves *pax6a/b* function. We utilised the presence of duplicated *pax6a/b* genes in the zebrafish by studying how ASD develops in *pax6a/b* homozygous and compound hetero-/homozygous mutants through live multidimensional imaging of ocular NC cells and the corneal endothelium. Analysis of NC migration in wildtype and *pax6a/b* single, double and compound mutants revealed two ocular NC populations and a role for Pax6 in correctly guiding their migration. Expression of guidance molecules in the optic cup and its surroundings including NC cells was altered in the *pax6a/b* double homozygous mutant. Unlike in murine Pax6 mutants, the proximodistal patterning of the optic cup was normal. We demonstrate instead a role of Pax6 in the control of the expression of NC guidance molecules, uncovering a so far unrecognised late function of Pax6 independent of the earlier proximodistal

patterning of the optic cup. The double homozygous mutant failed to establish the lens, leading to severe AS defects, with no formation of the corneal endothelium. Similar AS defects were observed in wild type embryos after chemical inhibition of TGFβ signalling or removal of the lens. Conversely, transplantation of a wildtype lens into the double homozygous mutant eye partially restored the AS. The ASD with misguided NC differentiation in *pax6a/b* mutants underscores the role of the second NC wave in governing distal eye maturation.

## Results

### Two distinct populations of cranial NC cells contribute to the developing eye

To trace the NC cells entering the eye, two transgenic lines were generated which express the green-to-red photoconvertible Eos fluorescent protein (EosFP [17]) in NC cells under control of *sox10* regulatory sequences. *Tg(sox10:mem-tdEosFP)* expresses a membrane-targeted construct of tandem-dimeric EosFP [18] and *Tg(sox10:h2a-tdEosFP)* directs tdEosFP to the nucleus. Expression of EosFP became first detectable between 11 and 12 h post-fertilisation (hpf) as two bilateral stripes of NC cells at the lateral border of the anterior neural keel (S1 Video). Between two lateral NC stripes, a temporal medially-located NC cell group was observed during 13–15 hpf. The anterior-most cells of the lateral stripes reached the proximal region of the nascent eye at 14 hpf (S1 Video) and started to envelop the proximal side of the eye (Fig 1A and 1E), eventually forming a monolayer by 18 hpf (Fig 1B and 1F; S1 Fig). These NC cells retain large cell-cell contact planes while enwrapping the proximal side of the optic cup and display a cuboidal cell shape (S1D–S1F Fig). They migrated as bilateral stripes (S1 Video). Importantly, this group of NC cells remained restricted to the proximal half of the eye and did not settle in the distal AS (S1 Fig). In this study, we refer to this early eye-colonizing NC cells as 1˚NC cells in contrast to a second group of NC cells, newly defined in this study, which we term 2˚NC cells.

These 2˚NC cells approach the eye later from the dorsal side at 20–22 hpf (magenta circles in Fig 1C and 1G). At 17–18 hpf, we observed that 2˚NC cells formed a transient cell cluster near the dorsal edge of the eye (magenta circles, Fig 1B and 1F; S1 Video). Subsequently around 22 hpf, the 2˚NC cluster dispersed into individual cells with a mesenchymal morphology and migrated over the dorsal rim of the eye into the distal region (S2 Video). In contrast to 1˚NC cells which migrated in groups with large cell-cell contacts (S1D–S1F Fig), 2˚NC migrated as single cells with maximally filopodia-like contacts to neighbouring migrating cells (Fig 1I). At 40 hpf, Sox10 reporter-positive NC cells were found to contribute to the distal and periocular tissues of the eye (Fig 1J–1J").

### Sox10 transcription becomes inactivated after dissolution of the 2˚NC cell cluster

At the 40 hpf stage, 2˚NC cell derivatives retained only very low levels of expression of the Sox10 reporter and we could not continue to trace their fate in AS structures, suggesting that *sox10* expression in 2˚NC cells has been turned off by this stage. In agreement, upon migration of 2˚NC into the eye, *sox10* mRNA expression was detectable only in a few mesenchymal cells in the distal side of the eye, suggesting that *sox10* is down-regulated in NC cells once they have entered the differentiation phase [19], in contrast to those of the other eye mesenchymal marker genes *pitx2* and *foxc1a* (S2 Fig). Similar observations were made with the 1˚NC cells except that we failed to detect mRNA expression earlier when we examined the RNA expression in these NC cell derivatives in the proximal eye at 18 hpf.

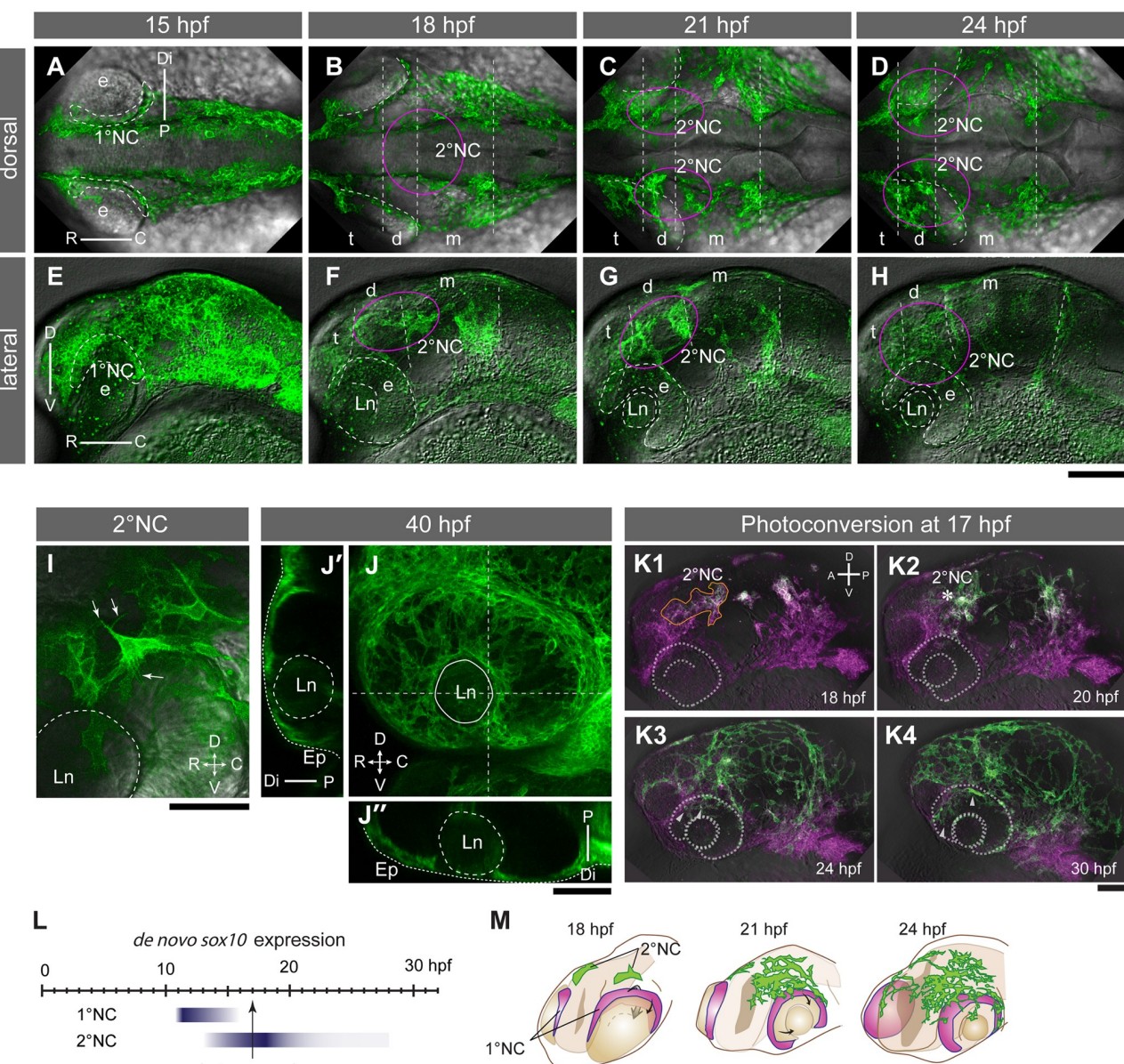

**Fig 1. Development of the AS of the eye visualized by *Tg(sox10:mem-tdEosFP)* and *Tg(corneaEndo:GFP)*.** (A-J") Confocal fluorescent images of *Tg (sox10:mem-tdEosFP)*-NC cells (green). Panels A-H represent time series of embryos viewed dorsally (A-D) and laterally (E-H) at 15- (A, E), 18- (B, F), 21- (C, G) and 24 hpf (D, H). (A, E) 1°NC cells (dotted line) spread over the proximal surface of the eye (e). Orientations: proximal (P), distal (Di), rostral (R), caudal (C), dorsal (D), ventral (V). (B, C, F and G) Clusters of 2°NC cells (within the magenta circle) are observed next to the diencephalon (d) and mesencephalon (m) at 18 hpf. t: telencephalon; Ln: lens. (D, H) 2°NC clusters dissolve and individual cells migrate into the distal side of the eye. (I) 2°NC cells that migrate into the distal side of the eye exhibit mesenchymal morphology with filopodia. (J-J") At 40 hpf, the nascent AS of the eye harbours mem-tdEosFP-positive NC cells. Transverse (J') and horizontal (J") optical sections along the dashed lines through the lens show the localization of NC cells beneath the surface epithelium (Ep, stippled line). Scale bars: A-H, 100 μm; I-K4, 50 μm. (K1-4) Photoconversion of *Tg(sox10: mem-tdEosFP)* embryos at 17 hpf visualizes 1°NCs and 2°NCs separately in magenta and green, respectively. Lateral views of time lapse fluorescent images were merged over the bright field images. The eye primordium and lens are indicated by dotted lines. Note that 2°NC cells (orange line, K1) continue to express *sox10* reporter, resulting in a white colour in the presence of photoconverted mem-tdEosFP (magenta). 2°NC cells located at the dorsal edge of the eye primordium (asterisk, K2) migrate into the distal eye compartment (arrowheads, K3-4). (L) Summary of the period of active *sox10* expression in 1° and 2°NC cells elucidated by serial photoconversion experiments (S2 Fig). (M) Schematics showing the position of 1°NC (magenta) and 2°NC cells (green) at 18, 21 and 24 hpf.

By using the green-to-red photoconversion property of EosFP upon exposure to 400 nm light [20], we estimated more precisely when *sox10* expression is silenced. The green form of the EosFP reporter in *Tg(sox10:mem-tdEosFP)* embryos was converted to the red form at different time points from 16 to 24 hpf and the recovery of green fluorescence was monitored at 36 hpf (S2 Fig). When photoconverted at 16 hpf, the 1°NC cells did not recover green EosFP (S2 Fig, asterisks in D'), suggesting that the 1°NC had stopped to express the *sox10* reporter already as early as 16 hpf (S2 Fig; summarized in Fig 1L). In contrast, the 2°NC cells recovered green EosFP expression when converted during 16–20 hpf, but not beyond 20 hpf (S2 Fig). This time point coincided with the dissolution of the 2°NC cell cluster and the arrival of individual 2°NC cells at the distal eye segment. Photoconversion at 17 hpf is thus an effective way to visually distinguish between 1°NC and 2°NC populations (Fig 1K1-4 and 1M).

## Wnt reporter is consecutively activated in 1° and 2°NC

Previously, Wnt signalling has been implicated in the control of NC cell differentiation [21]. To assess canonical Wnt signalling activity in 1° and 2°NC cells, *Tg(7xTCF-siam:nls-mCherry)* [22], in short *Tg(Wnt-rep)*, was crossed into the NC reporter line *Tg(sox10:h2a-tdEosFP)* and imaged by using a custom-built, multi-channel, high-resolution digital scanned laser light sheet fluorescence microscope (DSLM) [23]. During 13–15 hpf, transient expression of the Wnt reporter was observed in NC cells; expression became barely detectable by 15 hpf (Fig 2A–2C' and 2F; S3 Video). Starting at 18 hpf, we found intense Wnt reporter activation in the 2°NC cells (Fig 2D–2E' and 2F; S3 Video). This second activation of the Wnt reporter in 2°NC cells occurred in approximately 50% of NC cells and was maintained after entry of 2°NC cells into the distal side of the eye (Fig 2G, S3 Video, S3I Fig). Thus, 2°NC cells represent a heterogeneous group of cells with respect to Wnt activation.

We next tested whether Wnt signalling plays a role in 2°NC specification, migration or differentiation. Treatment of embryos with the canonical Wnt inhibitor IWR-1-endo [24] showed a reduction of *sox10*-expressing cranial NC cells (S3D Fig). However, neither migration of 2°NC cells into the eye nor formation of a coherent endothelial layer were affected by the treatment (S3 Fig). Irrespective of the function of Wnt signalling in 2°NC, the combination of Wnt reporter and *sox10* gene expression provides an unequivocal marker for a subpopulation of 2°NC cells *in situ*.

## NC cells destined for the distal side of the eye originate from the medial dorsal neural keel

Next, we assessed the relationship between the origin and destination of the two NC cell populations contributing to the eye quantitatively. To this end, we imaged *Tg(sox10:h2a-tdEosFP)* embryos by DSLM and employed our recently developed interactive image analysis framework [25] to trace back the origin of NC cells that arrived at the proximal and distal sides of the eye (S4 Video). NC cells for each destination were selected at two stages (17 and 24 hpf) for precise group selection based on the final location (Fig 3A). Since 1°NC and 2°NC cell populations are well separated at 17 hpf (Fig 1L), we combined a forward and backward strategy to track cells of the proximal and distal populations after continuous recordings from 11 to 31 hpf (Fig 3A). By backward and forward tracking of selected nuclei, we established the route, origin and destination of individual cells of proximal and distal NC populations (*n* = 2 embryos; S4 Fig).

For the analysed cells, we compiled the results in tracking diagrams by superimposing the backward and forward tracks of selected cells covering developmental time intervals ranging from 17 to 11 hpf and 17 to 31 hpf, respectively (Fig 3B–3E', S5 Video). This in-depth analysis confirmed that the cells occupying the proximal side of the eye were all formed at 12–14 hpf in

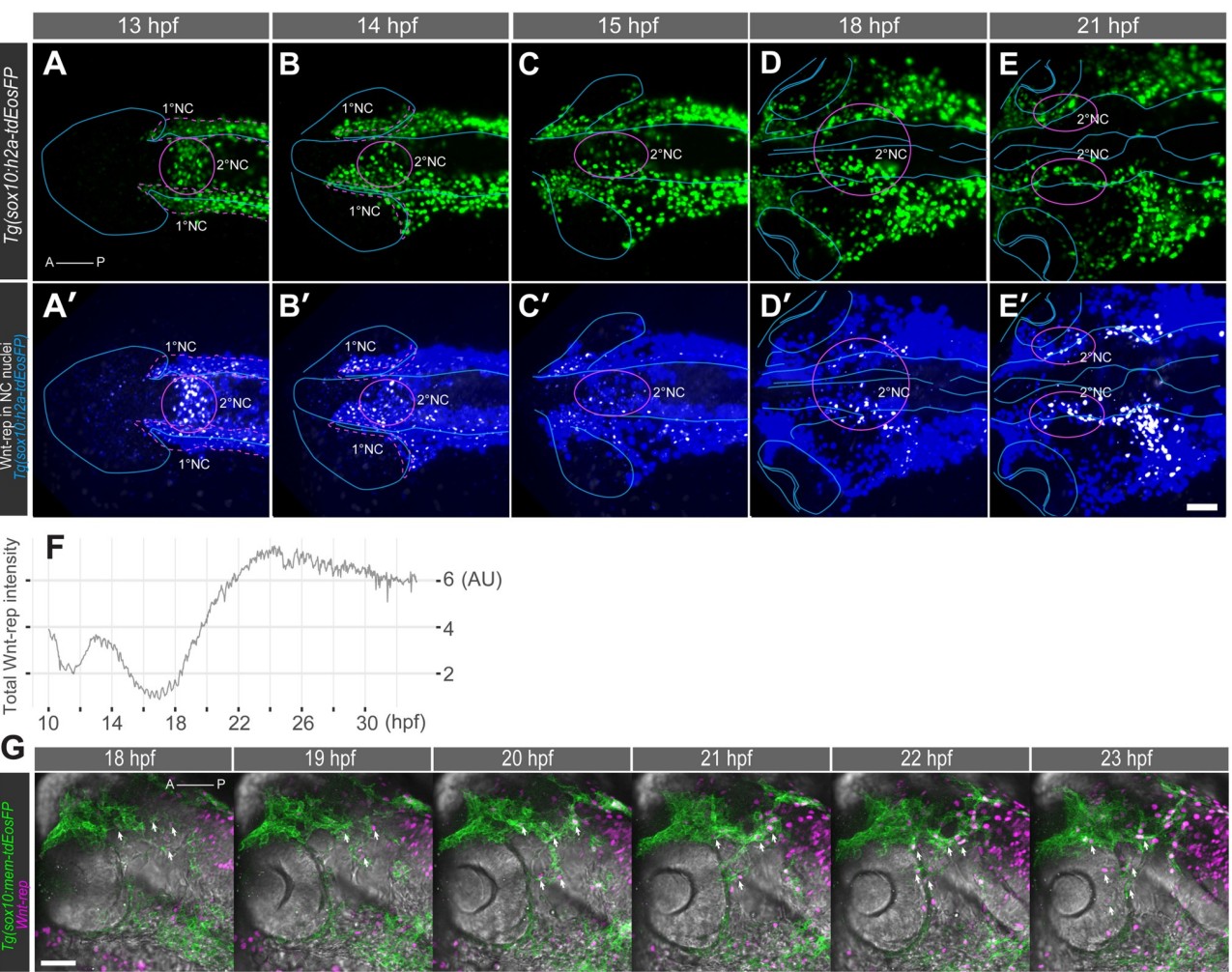

**Fig 2. Two waves of Wnt reporter expression.** Dorsal views of a *Tg(sox10:h2a-tdEosFP;wnt-rep)* embryo highlighting (A-E) h2a-tdEosFP-positive NC cells and (A'-E') Wnt reporter expression (white) in h2a-tdEosFP-positive NC cells (blue) showing colocalized Wnt reporter with nuclei from *sox10*-positive NC cells. At 13–14 hpf, 2°NC cells in the medial region (magenta circles, A-B) express Wnt-rep (A'-B') at higher levels than 1°NC cells. This Wnt-rep expression is transient and followed by a reduction at 15 hpf (C'). At 18–21 hpf, 2°NC cells develop into bilateral clusters at the level of the diencephalon and mesencephalon (magenta circles, D-E). These cells express high levels of Wnt-rep (D'-E'). Orientation: anterior left. (F) Mean total intensity (expressed in arbitrary unit, AU) profile of Wnt reporter expression in *sox10*-NC cells during 10–33 hpf shows 1st and 2nd wave of Wnt activity. (G) 2°NC cells (green) express Wnt reporter (magenta) prior to and during migration into the distal side of the eye (arrows). Orientation: anterior left, dorsal up; Scale bars: 50 μm.

the lateral region of the neural keel (Fig 3B–3B', S4A and S4B Fig), and are thus derivatives of 1°NC cells. The NC cells destined for the distal side of the eye mainly originated medially near the diencephalon and mesencephalon at 16–17 hpf (asterisk, Fig 3C), with a minor contribution of NC cells that originated in the lateral regions of the neural anlage (S5 Fig). Forward tracking showed that distal NC cells reached the eye via the dorsal rim of the optic cup (arrow, Fig 3E'). The lineage tree analysis on manually corrected data from an embryo identified comparable lineage sizes for NC groups destined to the proximal (*n* = 57 lineages) and distal eye (*n* = 46 lineages), respectively (Fig 3F and S7 Fig). Most of the analysed lineages (*n* = 113; 91%) produced a homogeneous NC population that exclusively consisted of NC cell derivatives in either the proximal or distal eye, showing that these NC cells with different destinations mostly belong to distinct lineages. Ten cells (9%) displayed a mixed lineage suggesting that NC cells

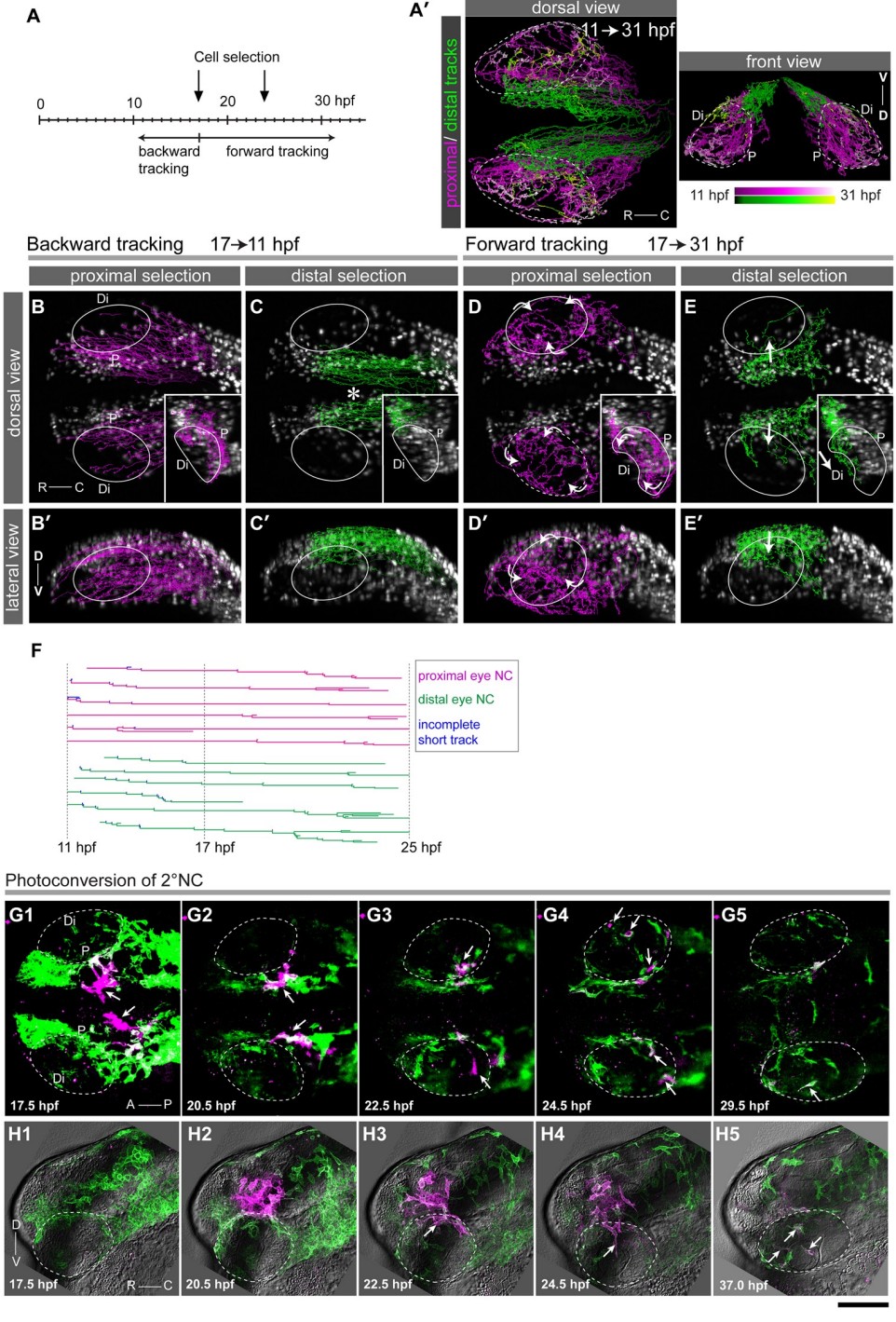

**Fig 3. Systematic tracking of 1°NC and 2°NC cells.** DSLM time series of a double transgenic embryo expressing nuclear Eos in NC cells [green, *Tg(sox10:h2a-tdEosFP)*] and nuclear mCherry from the Wnt reporter. (A-F) Tracking analysis of 1°NC and 2°NC cells. Two groups of *h2a-tdEosFP*-positive NC cells which were located at either the proximal or distal side of the optic cup were selected at 17 and 24 hpf for systematic backward (B-C') and forward tracking (D-E'), relative to 17 hpf (A). (A') Dorsal and front views of trajectories of NC cells destined for the proximal (magenta, 474 cells) or the distal side (green, 610 cells) of the eye are presented with colour-codes for approximate time periods. Stippled circles represent the optic cup. B-E, dorsal projection view; B'-E', lateral projection. With reference to the time of cell selection (17 hpf), backward (17–11 hpf, B-C') and forward tracking (17–31 hpf, D-E') results are shown. Individual tracks are colour-coded for stages to indicate the temporal cell position (A). Tracks are merged onto the maximum projections at the end of the tracking period (17 hpf, B-C; 31 hpf, D-E; see also S5 Video for three-

dimensional presentation). 2˚NC cells at the dorsal edge of the eye originate from the diencephalon and mesencephalon (asterisk, C) and migrate into the distal side of the eye (arrows in E) from the dorsal edge (arrow in E'). Circles in the panels outline the eye. Insets in B-E show transverse sections made at the lens highlighting cell migration paths relative to the proximal (P) and distal (Di) side of the eye. Orientations: rostral (R), caudal (C), dorsal (D), ventral (V). (F) Depiction of six representative lineages for each of the proximal (magenta) or distal (green) destined group. Each branch corresponds to a cell division. Incomplete short tracks are shown in blue. (G-H) Time lapse analysis of locally green-to-red photoconverted 2˚NC cells at 17 hpf (G) and 20 hpf (H). Arrows indicate 2˚NC cells reaching the distal side of the eye. G, dorsal projection view; H, lateral projection. Scale bar: B-E': 100 μm; G-H: 75 μm.

with different destination are not totally distinct in their potential to migrate to and differentiate into AS structures. The tracking analysis furthermore confirmed the predominant migration of mediodorsally formed NC cells at 16–17 hpf into the distal side of the eye (82%, $n = 39$ NC cells), which was further supported by local photoconversion and time lapse analysis (Fig 3G–3H; S6 Video). In contrast, 1˚NC cells rarely ended up in the distal part of the eye (3.6%, $n = 112$ NC cells). Thus, the destinations of NC cell groups with different origins are predominantly distinct; laterally located early migrating 1˚NC cells are destined for the proximal side of the eye, whereas medially located 2˚NC cells colonize predominantly the distal eye.

We next assessed whether 1˚NC derivatives residing in the proximal eye could contribute to distal structures at later stages by assessing the orientation of migration after 22 hpf, systematically. The orientations of migration of most of the NC derivatives in the proximal eye after 22 hpf (S6A and S6C Fig) do not support the notion that 1˚NC derivatives (S6E, S6F, S6H and S6I Fig) contribute largely to the distal segment of the eye. In contrast, the migration of the 2˚NC cells (S6G, S6J and S6K Fig) is predominantly oriented along the proximodistal axis with long tracks heading to the distal segment (S6B and S6D Fig).

## The 2˚NC cells arise at the dorsal diencephalon and mesencephalon

DSLM imaging lacks the bright field view for precise anatomical registration of the origin of 2˚NC. We thus mapped the origin of 2˚NC cells relative to those of the expression domains of known marker genes in the diencephalon and midbrain. At 18 hpf, *sox10* and Wnt reporter transcripts showed extensive colocalization in the two bilateral clusters of 2˚NC cells (Fig 4A–4A"). In contrast, transcripts of *sox9b* whose expression was suggested to mark pre-migratory NC cells [26] and the Wnt reporter did not colocalize (Fig 4B–4B"). Only very few cells expressed *sox9b* and *sox10* at the boundary between the expression domains of the two genes (Fig 4C–4C"). Thus, 2˚NC cells expressing *sox10* and Wnt reporter transcripts are located next to *sox9b* expressing diencephalic and mesencephalic regions. *sox10* did not show extensive colocalization with *pax6b* (Fig 4D–4E; S7A and S7B Fig), which is expressed similarly as *pax6a* in the presumptive thalamic portion of the diencephalon at 18 hpf [27]. *sox9b* showed almost complete colocalization with *otx5*, a marker of pineal photo-receptor cells (Fig 4F; S7C Fig) [28] in the dorsal-most region of the diencephalon, where *pax6b* is not expressed (Fig 4E'–4F'). Taken together, the 2˚NC cells arise from the dorsal mesencephalon and diencephalon close to the pineal anlage (Fig 4G–4I).

## Endothelial cells emerge in a concentric ring pattern in the nascent cornea

The corneal endothelium is derived from NC cells [3] and its establishment is a hallmark of the formation of the anterior chamber. We found that the *Tg(-3.1mnx1.1:GFP)^{ml4}* transgenic line [29] expresses GFP in the corneal endothelium. Although the DNA construct was originally developed to study spinal motor neurons, this particular insertion, which we mapped to the minus strand at the end of chromosome 9 (S9 Fig), is ectopically expressed in the corneal endothelium (Fig 5C–5C'"). We employed this line, which we call *Tg(corneaEndo:GFP)*, for

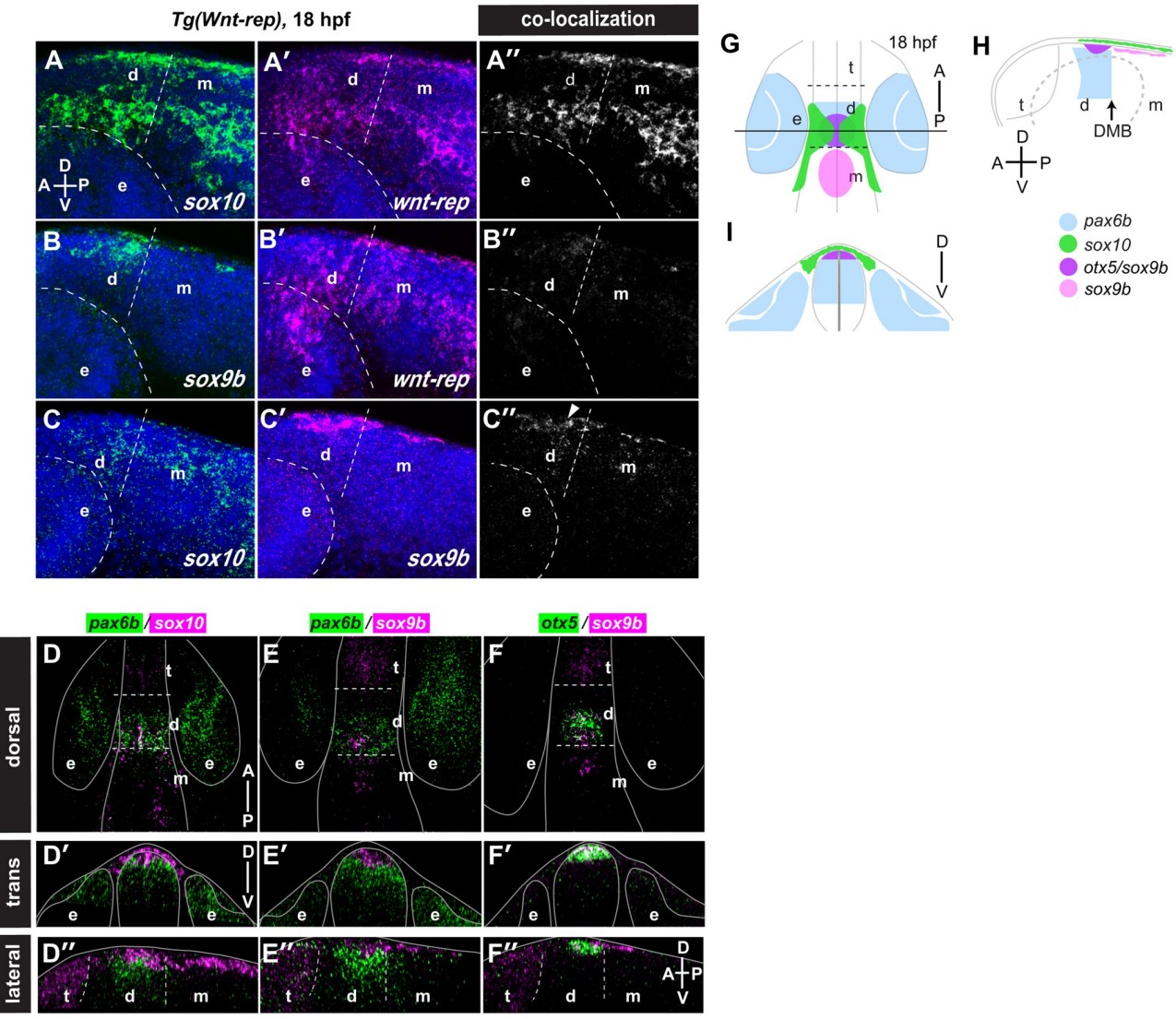

**Fig 4. 2°NC cells arise from the dorsal diencephalon and mesencephalon.** (A-C") Double *in situ* hybridisation analysis in the diencephalon of *Tg (Wnt-rep)* transgenic embryos at 18 hpf (lateral view, anterior left, dorsal up). Three combinations of probes were examined: *sox10/wnt-rep* (A, A'), *sox9b/wnt-rep* (B, B') and *sox10/sox9b* (C, C'). Panels A"-C" show co-localization of two probes as white region. Embryos were counterstained with DAPI to visualize nuclei (blue). Marginal co-localization of *sox9b* and *sox10* transcripts is indicated by an arrowhead (C"). (D-F") Pairwise comparisons of the expression patterns of *pax6b/sox10* (D-D"), *pax6b/sox9b* (E-E") and *otx5/sox9b* (F-F") are summarised in dorsal maximum projection views (D-F), transverse sections at the level of the pineal anlage (D'-F') and sagittal optical sections along the midline (D"-F"). At 18 hpf, *pax6b* and *sox10* expression (D) as well as *pax6b* and *sox9b* (E), appear to overlap when fluorescent signals were projected in dorsal views. Expression is, however, mostly confined to distinct dorsoventral layers (D'-E'). (F-F") *sox9b* co-localizes with *otx5*. (G-I) Summary of the gene expression pattern in dorsal view (G), in transverse view (I) along the horizontal line in G and in lateral view (H). The *sox10*-positive NC cells (green) reside next to *sox9b*- and *otx5* double positive cells of the epiphysis anlage (purple). t: telencephalon; d: diencephalon; m: mesencephalon; e: eye; DMB: diencephalon midbrain boundary; Scale bar: A-C": 100 μm; D-F": 150 μm.

live imaging. The first corneal endothelial cells emerge as a patch of less than 10 cells on top of the lens around the 60–72 hpf stage (Fig 5A). Corneal endothelial cells proliferate until they occupy the entire inner surface of the cornea (Fig 5B and 5C). The total number of corneal endothelial cells is $27 \pm 4$ cells ($n = 47$ embryos) at 120 hpf.

Based on this imaging analysis and the other data presented so far, we propose a model of five developmental phases of AS formation (Fig 5D). After formation of the 2°NC cell cluster

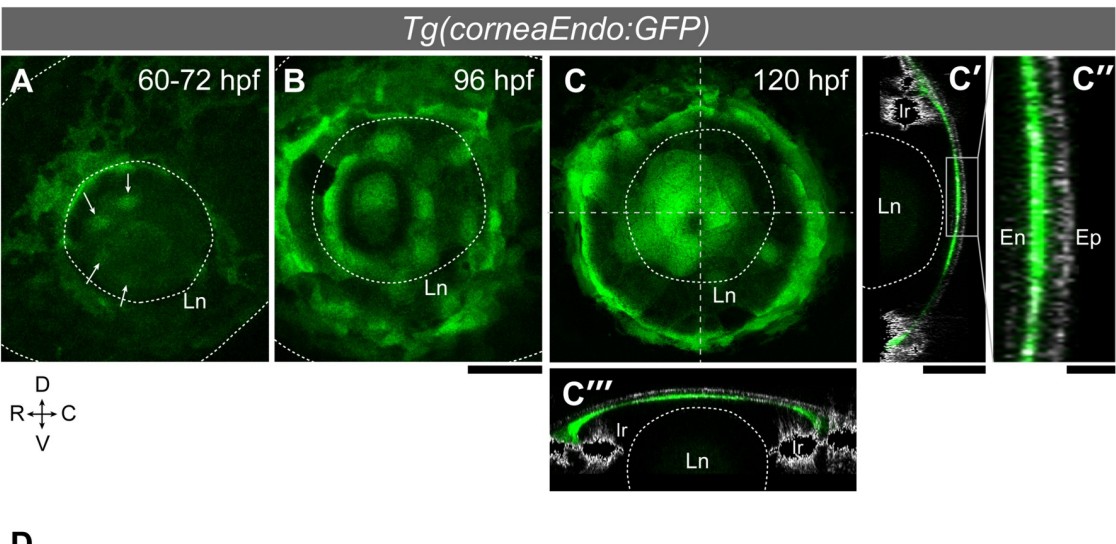

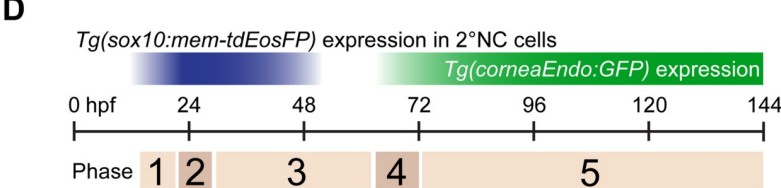

**Fig 5. Endothelial cells emerge in a concentric ring pattern in the nascent cornea.** (A-C") The *Tg(corneaEndo:GFP)* line shows the first corneal endothelium cells (green) as a patch of cells over the lens (arrows in A, 60–72 hpf) that later spread across the entire inner surface of the cornea (B: 96 hpf; C:120 hpf). (C'-C"') Transverse (C') and horizontal (C"') optical sections along the stippled lines in C, merged onto pictures from reflection imaging (grey channel) which highlight the corneal epithelium (Ep; note: the strong lateral signals are iridophores). (C") Magnified view of the central cornea region indicated by a rectangle in C'. En: corneal endothelium; Ln: lens; Ir: iris; Scale bars: 50 μm (A-C, C' and C"'); 10 μm (C"). (D) The AS development is classified into five phases. Phase 1: 2˚NC cell cluster formation (18–22 hpf); Phase 2: 2˚NC cluster dissolves and migrate into the eye (22–28 hpf); Phase 3: NC cells differentiate into ocular mesenchymal cells (28–60 hpf); Phase 4: First corneal endothelial cells emerge over the lens (60–72 hpf); Phase 5: Corneal endothelium fully covers the eye (>72 hpf).

at the diencephalon/midbrain level (Phase 1, 18–22 hpf), between 20–30 2˚NC cells migrate into the distal side of the eye (Phase 2, 22–28 hpf), attenuating expression of the NC marker *sox10*. The 2˚NC cells differentiate into ocular mesenchymal cells (Phase 3, 28–60 hpf). The first corneal endothelial cells emerge on top of the lens around the 60–72 hpf stage (Phase 4, Fig 5A). The number of corneal endothelial cells increased and they eventually occupied the entire inner surface of the cornea (Phase 5, >72 hpf, Fig 5B and 5C).

It was unclear where *sox10* is required for corneal endothelium differentiation. We thus analysed the structure of the cornea of homozygous *colourless* mutants (*sox10^{t3/t3}*) encoding a prematurely truncated Sox10 protein [30]. Whereas wild type or heterozygous siblings showed the normal structure of the cornea at 144 hpf (S10A nad S10B Fig), the homozygous *sox10* (*n* = 4) mutants showed irregularly shaped cells with large vesicular structures in the space between the lens and the stroma of the cornea (S10C and S10D Fig), indicating the requirement of Sox10 for proper corneal endothelium formation at this stage.

To assess whether the corneal marker *Tg(corneaEndo:GFP)* required Sox10 activity, a *sox10* morpholino directed against the *sox10* transcription start site [31] was injected into *Tg(corneaEndo:GFP)* zygotes. Expression of the corneal endothelial marker was assessed at 5 dpf. Control embryos injected with a control morpholino developed 28.6 ± 3.6 GFP-positive endothelial cells (*n* = 25), whereas *sox10* knockdown embryos show 12.8 ± 6.7 GFP-positive cells

lining the inner surface of the cornea ($n$ = 28; Welch Two sample $t$-test $p$-value = 1.3 x10$^{-13}$; S10G–S10I Fig). These cells lining the interior surface of the cornea appear more irregular as seen also in the electron micrographs of the mutant corneae. Like the sox10 mutant, *sox10* knock-down embryos have a smaller eye. The sox10 morphants showed a smaller anterior chamber (S10G'–S10H' Fig, stippled double-head arc with arrows) and annular ligament cells expanded toward the centre of the cornea (GFP-positive cells outside of the circle, AL; S10H Fig), which is normally confined to the peripheral regions (S10G Fig). These results are consistent with a requirement of Sox10 for proper development of the anterior chamber.

## Pax6a/b dose correlates with ASD severity

*pax6b* mutant zebrafish embryos develop ASD [16]. Zebrafish have duplicated Pax6 genes. Pax6a and Pax6b share 92.5% amino acid sequence identity and have overlapping patterns of expression [32]. To assess the impact of *pax6a/b* on development of the anterior chamber, we embarked on a single and compound mutant analysis. We utilised the previously identified ENU (*N*-ethyl-*N*-nitrosourea)-induced *pax6b* mutant *sunrise* (*tq253a* allele) [32]. A *pax6a* mutant line was created by deleting exons 8–12, the C-terminal half encoding the homeobox and proline/serine/threonine-rich (PST) domains using CRISPR/Cas9 gene editing (Fig 6; S11A–S11D Fig). We first analysed the AS phenotype of various *pax6a/b* mutant combinations at 5 dpf when the AS is fully established [14,15]. The *pax6a* zygotic homozygous mutant showed a marginal ASD at 5 dpf with a slightly smaller eye and lens, and an underdeveloped iridocorneal angle (Fig 6C–6C', arrows). In contrast, the *pax6b* maternal zygotic homozygous mutants presented a range of severe ASD classified into either keratoconus (Fig 6–6D') or hypotonic deflated anterior chamber often associated with corneal hyperplasia (Fig 6E–6E', arrow). The lens, smaller in size and occasionally with gaps between the epithelial cells and fibre cells (arrowheads, Fig 6D'–6E'), localizes at an abnormally distal position. Largely, AS phenotypes of *pax6a/b* single and double mutants supported an overall correlation between the degree of reduced *pax6a/b* gene copy number and the severity of ASD (Fig 6B–6I), as observed in mouse studies [11][33]. The distal side of the double mutant eyes was severely affected: it had no lens and was occupied with mesenchymal cells (Fig 6G and 6I, arrows; S7 Video), whereas the proximal side appeared less disturbed, with the exception of the overall smaller size of the eye. The corneal endothelium reporter was not detected in corneas of double mutants generated by injection of *pax6b-/-* embryos with *pax6a* gRNAs/Cas9 (Fig 6J–6K'; S8 Video). Thus, a differentiated endothelial layer has not been formed. *pax6a* gRNA-injected *pax6b-/-* embryos showed lensless eyes with 100% penetrance and were phenotypically comparable to *pax6a-/-;pax6b-/-* double homozygous mutants. The periocular mesenchymal cells observed in the AS of *pax6a* gRNA-injected *pax6b-/-* embryos retained abnormal expression of the mesenchymal marker gene *pitx2* (Fig 6L–6M) but lacked expression of the mature corneal epithelium marker *zgc:92380* (Fig 6N–6O) [16]. Taken together, this suggests that migration and differentiation of NC cells may not have occurred normally in the *pax6a/b* double mutants.

## Pax6a/b genes are required for correct migration of 1° and 2°NC

We next investigated the effects of Pax6 gene knockout on NC cell migration. We analysed *pax6a/b* compound mutants carrying the NC reporter line *Tg(sox10:h2a-tdEosFP)* in the background. Photoconversion of the EosFP reporter at 17 hpf to the red colour allowed unambiguous identification of 1°NC and 2°NC cells by their maintained red and recovered green fluorescence, respectively (Fig 7A'; S9 Video). Normal NC behaviour was observed with *pax6a +/+;pax6b+/+* (wild type), *pax6a+/+;pax6b+/-* (Fig 7A–7C; magenta and green lines for the

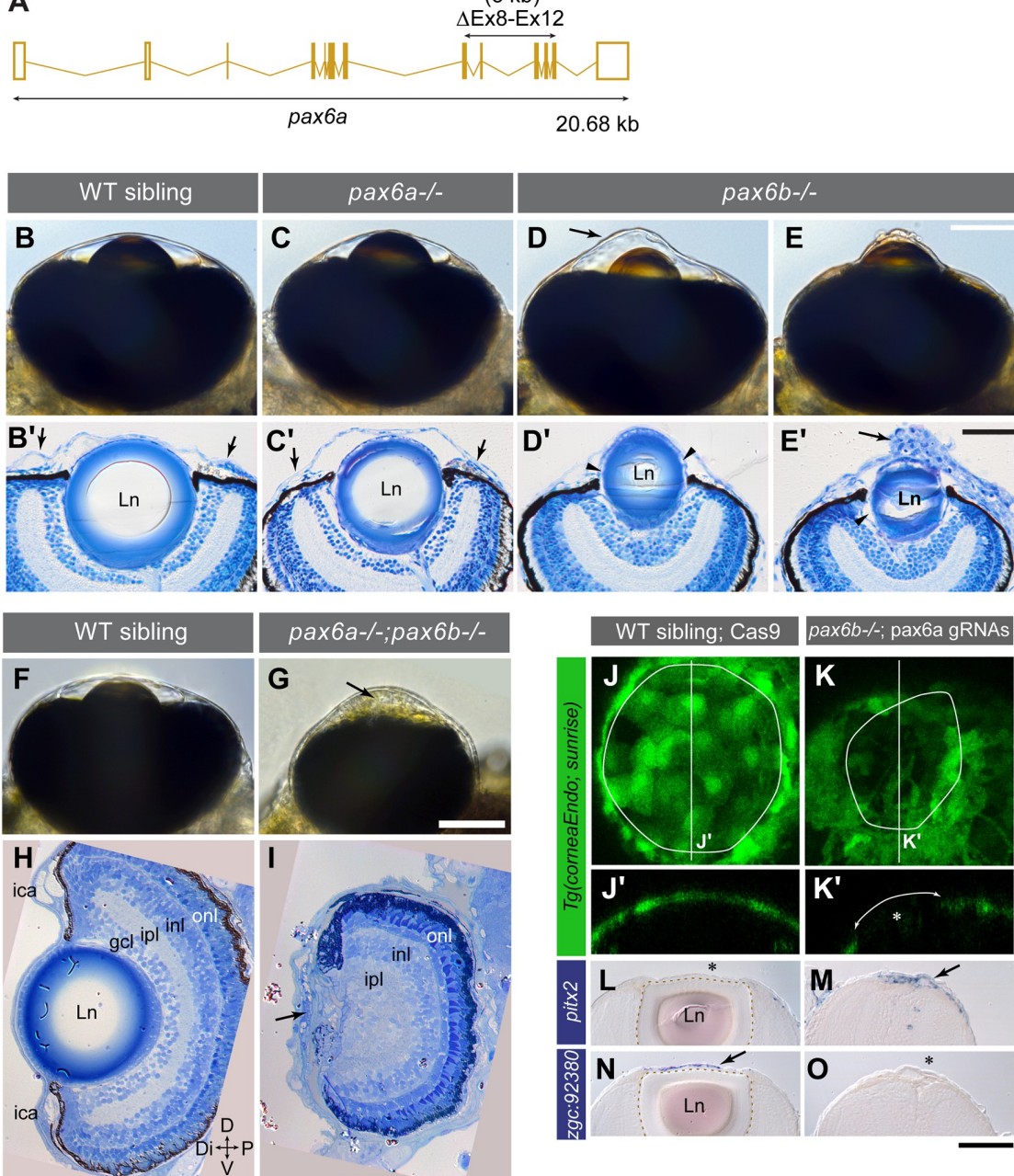

**Fig 6. Pax6a/b dose correlates with ASD severity.** (A) CRISPR/Cas9-mediated deletion of 3 kb in the *pax6a* locus encompassing exons 8–12 (ΔEx8-Ex12, a horizontal double head arrow) encoding homeobox DNA binding and PST domains (S11 Fig). (B-E') The AS phenotype at 5 dpf of a wild type sibling (B-B'), a *pax6a-/-* zygotic mutant (C-C') and *pax6b-/-* maternal zygotic mutants with keratoconus-like phenotype (D-D'; arrow in D) and deflated cornea (E-E'), imaged alive (B-E) and in sections (B'-E'). (B'-E') The chamber angle (arrows in B'-C') is open in wild type (B'), *pax6a-/-* (C') and *pax6b-/-* embryos with keratoconus (D'), however, it appears closed in *pax6b-/-* embryos with deflated cornea (E'). Although a *pax6a-/-* embryo appeared normal with live inspection (C), in comparison to the wild type that has a flat iris plane (B') the iris plane is slanted (C'), as also seen in *pax6b-/-* embryos (D'-E'). All *pax6* mutants show a smaller lens (Ln). (D'-E') Gaps are evident between the lens capsule and the lens fibre (arrowheads). Abnormal layering in the corneal epithelium is occasionally observed with deflated corneas (arrow in E'). (F-I) The AS phenotype at 5 dpf of a wild type sibling (F, H) and a double homozygous *pax6a-/-;pax6b-/-* embryo (G, I). ica: iridocorneal angle, gcl: ganglion cell layer, ipl: inner plexiform layer, inl: inner nuclear layer, onl: outer nuclear layer. The distal eye structure of *pax6a/b* double mutants has no lens, no gcl, no ica and is occupied with abnormal ocular mesenchymal cells (arrows in G, I). Orientation is dorsal (D) up, ventral (V) down, distal (Di) left and proximal (P) right. (J-O) Marker gene expression analysis in 5 dpf embryos of wild type siblings injected with Cas9 without gRNAs (J, J', L, and N) and of maternal zygotic *pax6b-/-* mutant genotype injected with *pax6a* gRNAs

and Cas9 (K, K', M and O). (J-K') The corneal endothelium visualized by the *Tg(corneaEndo:GFP)* transgenic line. The anterior chamber margin is demarcated by a solid circle. Optical sections made along vertical lines in J and K are shown in J' and K', respectively. The Pax6 deficient embryo (K-K') lacked both the lens (asterisk in K') and the expression of the corneal endothelium reporter (K', double arrowed arc). (L-O) *in situ* expression analysis of a mesenchymal marker *pitx2* (L-M) and of the corneal epithelium marker *zgc:92380* (N-O). Presence and absence of gene expression is denoted by arrow and asterisk, respectively. The lens border is demarcated by a stippled line. Scale bars: 50 μm.

1°NC and 2°NC trajectories, respectively) and *pax6a+/-;pax6b+/+* (see also Table 1). In contrast, in embryos with genotypes carrying a homozygous *pax6b* mutation, i.e., *pax6a+/+; pax6b-/-*, *pax6a+/-;pax6b-/-* and *pax6a-/-;pax6b-/-*, the 2°NC cell cluster did not dissolve but rather persisted at the distal-dorsal periocular site with high penetrance (Fig 7E–7E', solid circle; Table 1). As a result, these cells became excluded from the eye (Fig 7E–7E', stippled circle; S9 Video). Morpholino-mediated knockdown of *pax6a/b* mimicked this abnormal 2°NC cluster behaviour (S12 Fig). 2°NC cells failing to enter the eye maintain their *sox10* expression

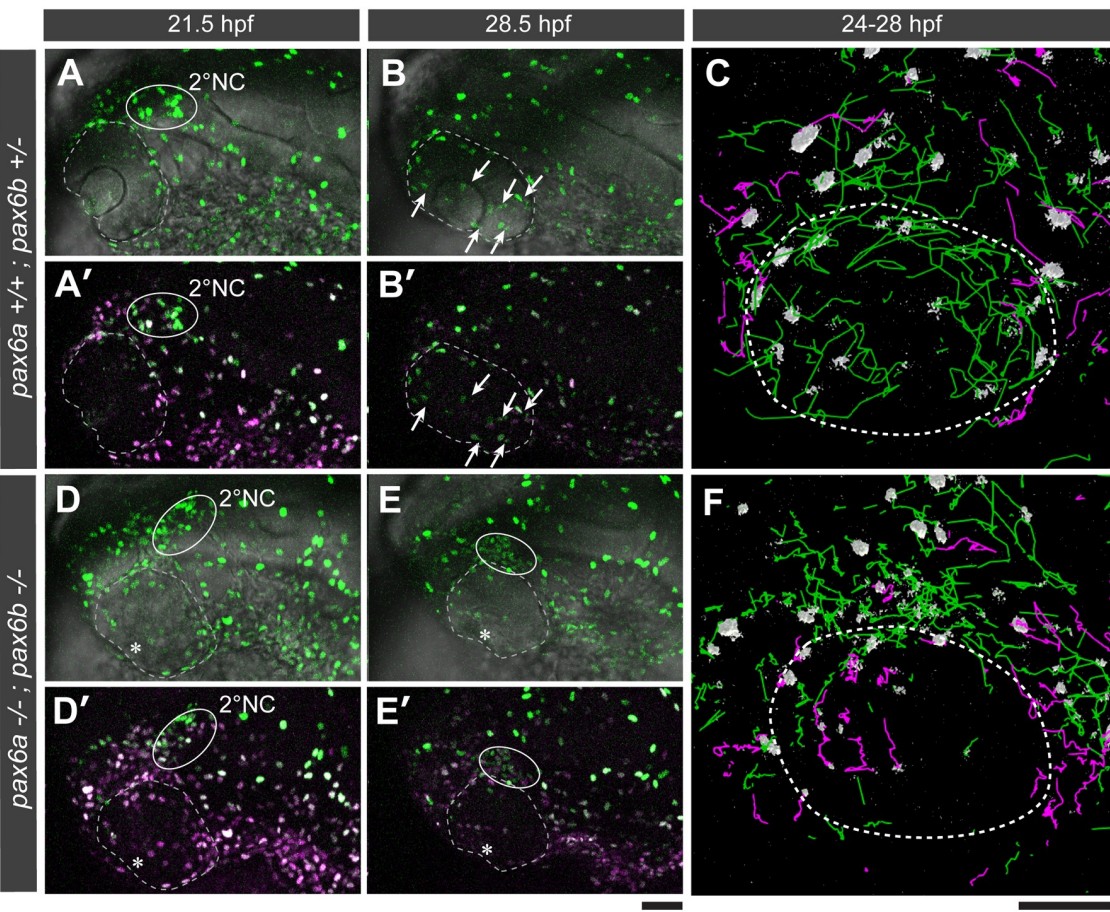

**Fig 7. Pax6a/b are required for guiding the two ocular NC populations to their distinct destinations.** *Tg(sox10:h2a-tdEosFP)* embryos with a genetic background of either *pax6a+/+;pax6b+/-* (A-C, wild type-like) or *pax6a-/-;pax6b-/-* (D-F, double homozygous mutant) were photoconverted at 17 hpf for visualizing 1° and 2° NC cells in magenta and green, respectively. At 21.5 hpf, the 2°NC cell cluster (A-A' and D-D'; solid circles) is observed at the dorsal edge of the eye in both genotypes. At 28.5 hpf, although 2°NC cluster was dissolved into individually migrating 2°NC cells in the distal eye compartment of a *pax6a+/+;pax6b+/-* embryo (B-B', arrows), the 2°NC cluster persisted at the same location in a *pax6a-/-;pax6b-/-* mutant embryo (E-E', solid circle) which failed to develop the lens primordium (D-E', asterisks). (C and F) Lateral views of 3D rendered trajectories of 1°NC (magenta lines) and 2°NC cells (green lines) for each genotype during 24–28 hpf. The NC nucleus is represented as a grey object. The eye boundary is shown as a stippled circle. Scale bars: 50 μm.

**Table 1.  *pax6a/b* compound mutant genotypes and NC migration defects.**

| Genotype | 2˚NC migration defect penetrance | AS phenotype |
|---|---|---|
| *pax6a+/+;pax6b+/+* | 0% (0/3) | normal |
| *pax6a+/+;pax6b+/-* | 0% (0/9) | normal |
| *pax6a-/-;pax6b+/+* | 0% (0/3) | normal (in case of zygotic mutants); severe defects (in case of *pax6a* maternal-zygotic mutants) |
| *pax6a-/-;pax6b+/-* | 0% (0/4) | n.d. |
| *pax6a+/-;pax6b+/+* | 25% (3/12) | n.d. |
| *pax6a+/-;pax6b+/-* | 39% (9/23) | n.d. |
| *pax6a+/+;pax6b-/-* | 100% (5/5) | severe defects (lens-cornea attachment, keratoconus) |
| *pax6a+/-;pax6b-/-* | 88% (7/8) | n.d. |
| *pax6a-/-;pax6b-/-* | 100% (3/3) | severe defects (no lens, no anterior chamber) with 100% penetrance |

*pax6a/b* gene dosage affects migration of both 1˚NC and 2˚-NC populations. A total of 70 *Tg(sox10:h2a-tdEosFP)* embryos carrying a compound *pax6a/b* mutation were analysed under a confocal microscope for NC migration during 17–30 hpf. The two NC populations were identified by green-to-red photoconversion of EosFP reporter at 17 hpf (see also Fig 7). After imaging analysis each individual embryo was genotyped. The abnormal NC migration is characterized by premature migration of 1˚NCs into the distal eye and/or absence of 2˚NCs into the distal eye during 17–30 hpf. The penetrance of the NC migration defect is given as the percentage of abnormal cases with respect to the total number of embryos in parenthesis. AS phenotype was evaluated under a dissection microscope at 5 dpf. n.d.: not determined.

(S12E and S12H Fig). In double heterozygous mutants, *pax6a+/-;pax6b+/-*, the 2˚NC entry into the distal eye compartment was affected in nearly half of the embryos (39%, Table 1), suggesting that this *pax6* gene number provides a borderline dose of Pax6 protein. In contrast to 2˚NC cells, 1˚NC cells were not affected except in the double homozygous *pax6a-/-;pax6b-/-* mutant: The 1˚NC cells migrated far into the distal half of the eye of these double mutants (Fig 7F, magenta tracks in the dashed circle). Taken together, our data are consistent with a gene number dependent effect of *pax6a/b* on the migration of the two distinct waves of NC cells.

## Pax6a/b genes are required for the expression of guidance molecules in the eye

Pax6 was proposed to convey positional information in the murine eye affecting the establishment of the three axis of the eye [34]. We thus examined markers indicative of the overall coordinates of the eye in *pax6a/b* compound mutants. For the sake of ease in crossing, we combined the zygotic *pax6a-/-* and the maternal zygotic *pax6b-/-* mutation for *pax6a-/-;pax6b-/-* double homozygous mutants. No change was observed between wild type, maternal zygotic *pax6b-/-*, and *pax6a-/-;pax6b-/-* double mutants with the dorsotemporal expression of the *aldehyde dehydrogenase 1 family member A2* (*aldh1a2*, arrows in Fig 8A–8A"), the dorsal expression of *T-box 5* (*tbx5*, arrows in Fig 8B–8B"), the proximoventral expression of *pax2a* (arrows in Fig 8D–8D"), the ventral expression of *ventral anterior homeobox 2* (*vax2*, Fig 8E–8E"), or nasotemporal expression of *foxg1a* and *foxd1* (S13 Fig). Also expression of *pax6a* and *pax6b* itself was not changed (Fig 8C–8C"; S11E and S11F"' Fig). As implied by the smaller size of the lens in maternal zygotic *pax6b-/-* mutants and its absence in *pax6a/b* double mutants, we observed a reduction of lens-specific expression of crystallin beta A4 (*cryba4*, arrows in Fig 8F–8F") with abnormal lenticular patterning (S11E and S11F" Fig). Thus, Pax6a/b proteins are required for lens primordium formation. However, they are not required for dorsal (defined

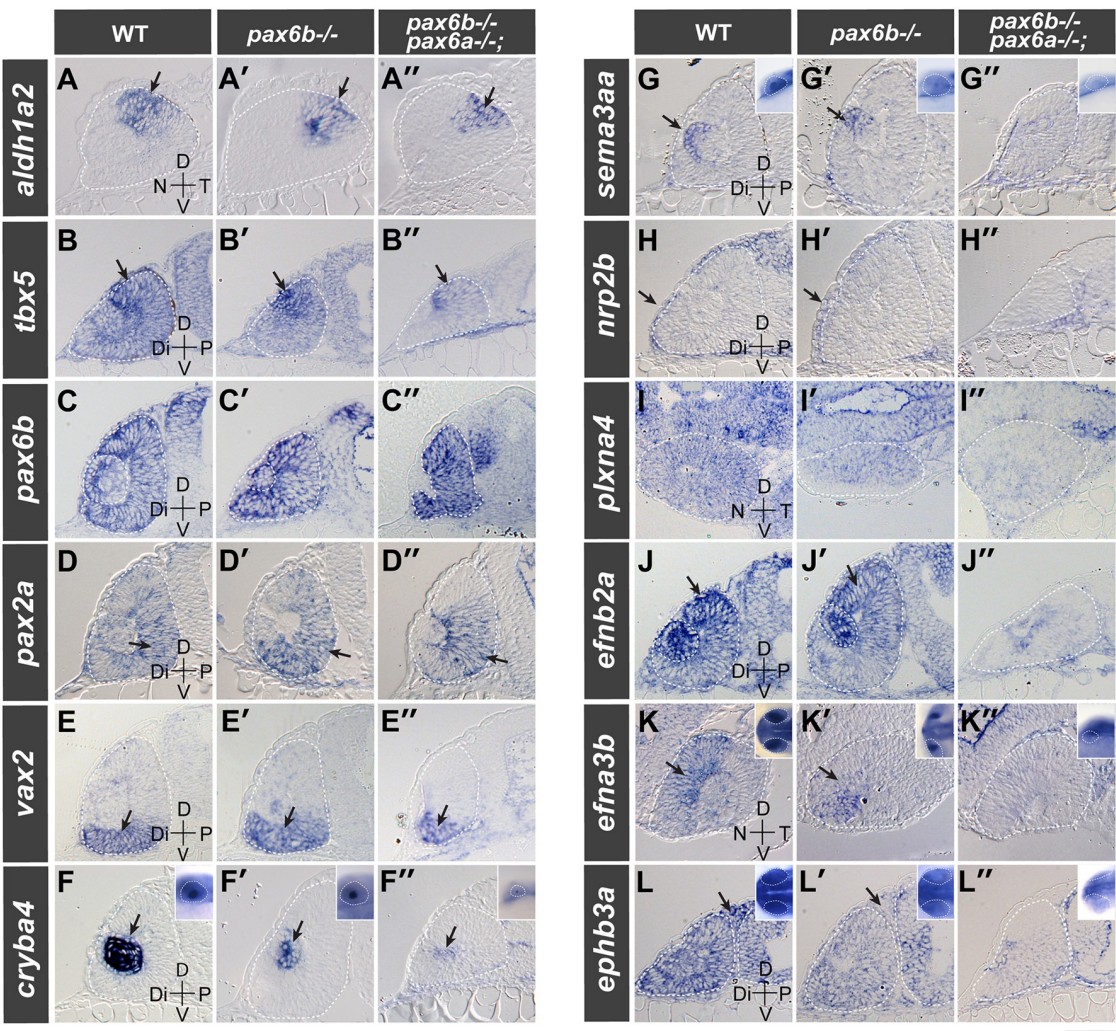

**Fig 8. Pax6a/b compound mutants lack expression of guidance molecules in the eye.** Gene expression was analysed by *in situ* hybridisation in wild type (WT, A-L), maternal zygotic *pax6b-/-* (A'-L') and *pax6a-/-;pax6b-/-* double homozygous mutant embryos (A"-L") at 28 hpf. The gene symbol of the transcript analysed is given at the left end of each row. The stippled white line demarcates the optic cup and brain. Sections are made either in transverse orientation (*tbx5* [B-B"], *pax6b* [C-C'], *pax2a* [D-D"], *vax2* [E-E"], *cryba4* [F-F"], *sema3aa* [G-G"], *nrp2b* [H-H"], *efnb2a* [J-J"] and *ephb3a* [L-L"]) or in sagittal orientation (*aldh1a2* [A-A"], *plxna4* [I-I"] and *efna3b* [K-K"]) to show the gene expression along the distal-proximal and nasal-temporal axis, respectively. For *cryba4* (F-F"), *sema3aa* (G-G"), *efna3b* (K-K") and *ephb3a* (L-L"), the head region of the whole mount embryos is shown in the inset. Note the expression indicated by the arrow: dorso-temporal for *aldh1a2* (A-A"), dorsal for *tbx5* (B-B"), ventro-proximal for *pax2a* (D-D"), ventral part of the optic cup for *vax2* (E-E"), lens primordium for *cryba4* (F-F") and *sema3aa* (G-G"), distal periocular mesenchymal cells for *nrp2b* (H-H"), dorsal for *efnb2a* (J-J'), dorso-nasal optic cup for *efna3b* (K-K") and periocular mesenchymal cells for *ephb3a* (L-L"). Scale bar: 50 μm; 200 μm for insets.

by *aldh1a2* and *tbx5*), ventral (*pax2a* and *vax2*), nasal (*foxg1a*), temporal (*aldh1a2* and *foxd1*) and proximal (*pax2a*) identity of the eye primordium.

Since the overall positional identities appear to be normal in the *pax6a/b* double mutant eye, we reasoned that *pax6a/b* may be required for correct expression of cell guidance molecules. In chick embryos, the expression of the soluble protein, semaphorin 3A in the lens epithelium is known to organize AS-contributing NCs by preventing them from prematurely migrating over the lens. [35]. While *semaphorin 3Aa* (*sema3aa*) expression was observed in the lens of *pax6b-/-* mutants (Fig 8G–8G'), its expression was absent in *pax6a/b* double

mutants (Fig 8G"). Semaphorin 3A is recognized by plexins and their co-receptors neuropilins [35]. *neuropilin 2b* (*nrp2b*)-positive ocular mesenchymal cells found in the distal end of the eye (Fig 8H) were present in *pax6b-/-* embryos (Fig 8H'). They were, however, absent in *pax6a/b* double mutants (Fig 8H"). The expression of the semaphorin receptor *plxna4* was reduced in the eye of both mutants (Fig 8I–8I"). We next examined ephrins and their cognate Eph receptors, both of which are involved in NC guidance. The dorsotemporal expression of the transmembrane ephrin ligand B2a [36] (*efnb2a*, arrows in Fig 8J) was detected in *pax6b-/-* mutants (Fig 8J') but not in *pax6a/b* double mutants (Fig 8J"). In contrast, dorsonasal expression of a glycosylphosphatidylinositol-anchored ephrin ligand, *efna3b* [37,38] (Fig 8K), and of Eph receptor B3a (*ephb3a*, Fig 8L) were reduced in *pax6b-/-* mutants (Fig 8K' and 8L'). Both markers were absent in *pax6a/b* double mutants (Fig 8K" and 8L"). In summary, Pax6a/b proteins are required for correct expression of guidance molecules in the ocular cup and its surroundings.

We hypothesized that this change in expression of guidance molecules affects the migration of the two NC cell populations. To test this notion, we injected morpholinos directed against the transcription start sites of *sema3aa* [39,40], *efnb2a* [41] and *nrp2b* [40], which are mainly expressed in the lens, lens/retina and NC cells, respectively, and monitored the movement of the NC cells. No significant effects on NC migration and AS formation were detectable upon knock-down of *sema3aa* and *efnb2a*. Abnormal NC migration and significantly reduced corneal endothelial cells were observed, however, when we knocked-down *nrp2b* (S14 Fig). The 1°NC cells of *nrp2b*-knockdown embryos migrated far into the distal half of the eye in comparison to the control knockdown (S14A Fig, $n = 14$ embryos), phenocopying the behaviour of 1°NC cells in *pax6b-/-;pax6a-/-* mutants (S14B Fig, arrow; $n = 17$ embryos). In contrast to the *pax6a/b* double mutant, 2°NC cells of *nrp2b*-knockdown embryos migrated normally into the anterior segment of the eye. Thus, altered *nrp2b* expression in NC cells of the mutants is only one aspect of the defects causing abnormal 1° and 2°NC migration. It also suggests that the two populations use different guidance cues. In accordance with the Pax6 double mutant phenotype the eyes were smaller. The endothelial layer was also reduced in number of cells with the characteristic flat morphology lining the inner surface of the cornea (Welch Two sample *t*-test *p*-value$<2.2$ x10$^{-16}$, S14E Fig). This reduction of endothelial cells, however, may reflect just the smaller size of the eye. The presence of the endothelial layer in the cornea is consistent with the migration of the 2°NC into the distal segment of the *nrpb2* morphants. Taken together, these results are consistent with the notion that the changes in guidance molecule expression in Pax6 mutants is an important part of the mutant phenotype.

## Transplantation of a lens into *pax6a/b* double mutants partially restores AS morphology

The lens has been proposed as an organising centre of the eye [42]. Pax6a/b double mutants lack the lens suggesting that the more severe defects of the double mutant may be linked to lack of the lens. We first characterized the defects of the anterior chamber of *pax6a/b* double mutants more closely. To visualize the size of the anterior chamber and the structure of the ocular blood vessels, we injected 2000 kDa FITC-dextran and 3 kDa rhodamine-dextran into the cardinal vein of 5 dpf larvae. The former dye remains inside the vessel while the latter diffuses across the vessel boundaries, allowing one to visualize the anterior chamber space. The *pax6a/b* double homozygous mutant has no anterior chamber and ocular superficial annular vessels were mislocalised (Fig 9D).

We next assessed the role of the lens in AS formation. When the lens was removed from the wild type eye at 24 hpf, the annular vessels were disorganised (Fig 9B), similar to the *pax6a/b*

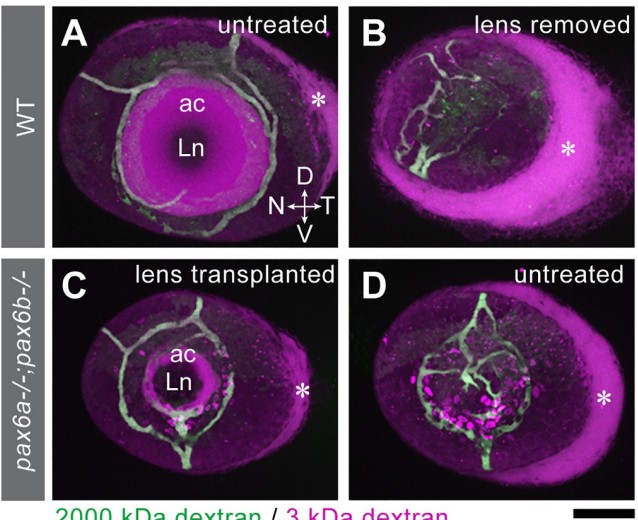

**Fig 9. Transplantation of a lens into *pax6a/b* double mutants partially restores AS morphology.** Intravenous injection of 2000 kDa dextran (green) and 3 kDa dextran (magenta) into wild type (A-B) and double homozygous *pax6a-/-;pax6b-/-* mutant embryos (C-D) at 5 dpf allows visualizing the anterior chamber (ac, magenta) with reference to ocular annular vessels (green) in en face view. The asterisk denotes diffusion of 3 kDa dextran dye through the uveoscleral pathway. (A) Wild type control anterior chamber with no surgical operation. (B) No anterior chamber was formed in wild type after lens removal at 24 hpf. (C-D) The anterior chamber was partially restored in *pax6a/b* double mutant eyes after transplanting a wild type lens at 24 hpf (C), while no anterior chamber was formed without operation (D). Orientations: dorsal (D), ventral (V), nasal (N) and temporal (T). Scale bar: 50 μm.

mutant eye (Fig 9D). To test whether the presence of a lens was sufficient to rescue AS formation in mutants, we performed lens transplantations [43]. When the wild type lens was transplanted into the eye of *pax6a/b* double homozygous mutants at 24 hpf, both the anterior chamber and the normal patterning of superficial annular vessels was restored (Fig 9C, *n* = 3 embryos). Thus, the absence of the lens in *pax6a/b* double homozygous mutants contributes to the ASD phenotype.

## TGFβ signalling is required for corneal endothelium formation

Lens-derived TGFβ2 organizes periocular mesenchyme differentiation to form the AS in mice [44]. Moreover, TGFβ2 was shown to down-regulate key periocular mesenchymal genes (including *Foxc1*, *Pitx2* and *Slug*) to initiate corneal endothelial differentiation *in vitro* [45].

We first analysed expression of the *tgfb2* gene (Fig 10A–10G). *tgfb2* gene was expressed at 22 hpf in the lens and also in mesenchymal cells found at the dorsal edge of the optic cup (arrowhead, Fig 10A). At 24 hpf, *tgfb2* mRNA was expressed in mesenchymal cells in both the distal and proximal side of the eye (arrowheads, Fig 10B). Interestingly, at stages after 28 hpf, *tgfb2* was not expressed in mesenchymal cells of the distal eye anymore (Fig 10C–10G). However, expression in the lens was detected until 48 hpf (Fig 10C–10G). Thus, *tgfb2* expression is highly dynamic.

We next examined the expression of the TGFβ reporter construct (Fig 10H–10N). The TGFβ signalling reporter line *Tg(smad3:GFP)* utilizes Smad3 binding elements to drive GFP expression [46]. TGFβ signalling dependent *gfp* transcription was first observed at 24 hpf in the dorsal side of the retina (arrow, Fig 10I) and in mesenchymal cells on the proximal side of the eye (arrowheads, Fig 10I). At 28 hpf, the TGFβ reporter is expressed in the lens (arrow, Fig 10J) and the retina, however, it is not observed in the distal side of periocular mesenchyme until 72 hpf. At 72 hpf, when distally located periocular mesenchymal cells start to differentiate

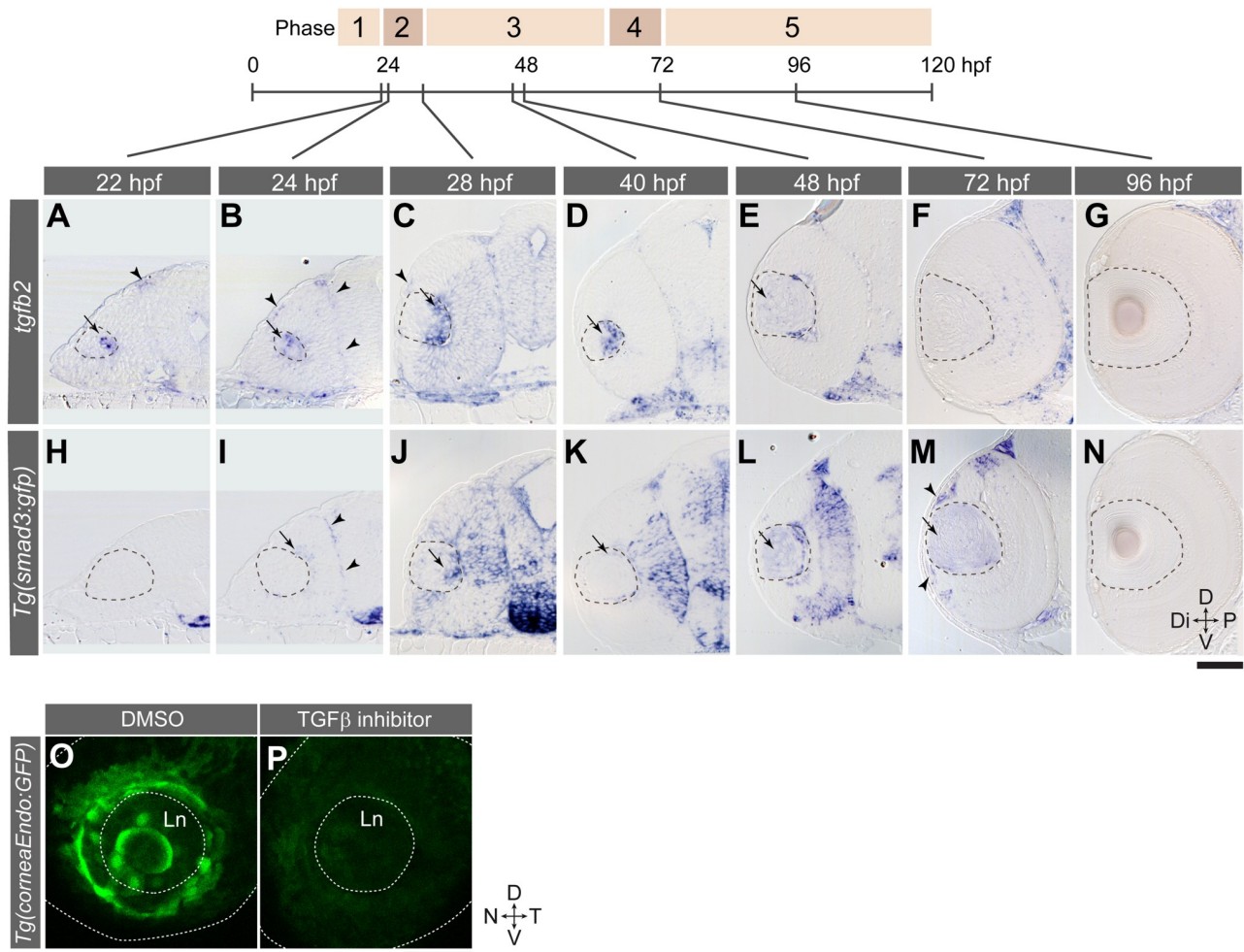

**Fig 10. TGFβ signalling in the lens is required for corneal endothelium formation.** (A-N) *Tg(smad3:GFP)* embryos were examined from 22–96 hpf for *in situ* expression of *tgfb2* (A-G) and *gfp* (H-N). The dotted line outlines the lens. Arrows and arrowheads indicate expression in the lens and mesenchymal cells, respectively. (O-P) *Tg(corneaEndo:GFP)* embryos were treated with either 1% DMSO (O) or TGFβ inhibitor (30 μM SB-505124; P) from 28 to72 hpf (stages 3–4). Orientations: dorsal (D), ventral (V), nasal (N) and temporal (T). Ln: lens. The dotted lines outline lens and eye cup. Scale bars: 50 μm.

into *Tg(corneaEndo:GFP)* positive cells (Fig 10O; Fig 5D), the TGFβ signalling reporter became expressed in the distally located mesenchymal cells (arrowheads, Fig 10M). This distal mesenchymal TGFβ signalling was preceded by activation of lenticular TGFβ signalling at 48 hpf (arrow, Fig 10L), which ceased at 96 hpf (Fig 10N).

Next, we examined whether inhibition of TGFβ signalling affects the development of the AS in zebrafish. During 46–87 hpf (stages 3–5 in Fig 5D), when NC cells start to differentiate, *Tg(corneaEndo:GFP)* embryos were treated with either 1% DMSO or the TGFβ signalling inhibitor SB-505124 at 30 μM. SB-505124 abolishes ALK4/5/7 kinase activity, thereby blocking Smad2/3 phosphorylation [47]. Reporter expression for corneal endothelial cells was abolished in treated embryos (Fig 10P, *n* = 4 embryos; S10 Video). Thus, TGFβ signalling is required for corneal endothelial differentiation. We did not observe prolongation of *sox10* expression in TGFβ inhibitor-treated embryos in contrast to the effects seen on mammalian NC cells *in vitro* [48].

## Discussion

Here we have identified two distinct groups of cranial NC cells that contribute to the development of the eye. The 1°NC cells, which envelop the proximal side of the eye, originate from the lateral premigratory NC cells. The 2°NC cells emigrate form more dorsomedial positions at the level of the diencephalon and mesencephalon a couple of hours later. They colonize the AS of the developing eye. Multidimensional live imaging revealed that 2°NC can be distinguished from 1°NC by their origin, their timing of delamination, target area in the eye, migration route, Wnt-reporter activity and maintenance of *sox10* expression. The entry of 2°NC into the eye is selectively blocked in *pax6b-/-* mutants. In *pax6a/b* double homozygous mutants, the behaviour of both NC populations was altered. The severity of ASD in *pax6* mutants appears to be correlated with the reduction of the *pax6* gene number. However, in contrast to the mouse, the proximodistal patterning of the eye was not affected in the zebrafish *pax6* mutants. NC migration phenotypes are correlated with lack or reduction of guidance molecules as well as the lack of the lens in the compound mutants. Severe ASD in *pax6a/b* double homozygous mutants can be alleviated by transplantation of a wild type lens which is, as in mammals, the source of TGFβ-like signals. In agreement, we found that chemical inhibition of TGFβ signalling inhibited differentiation of endothelial cells.

### Distinct waves of neural crest cells contribute to the morphogenesis of the eye

We traced the origin of the NC cells entering the eye to two distinct locations in the premigratory NC. The 1°NC cells colonising the proximal eye originate from the two lateral assemblies of premigratory NC cells described previously [21,49]. The 1°NC cells enter the eye posteriorly to cover the back side of the developing eye cup. The cells of the 2°NC group, which enter the eye dorsally and migrate to the distal side of the eye, originate from cells that reside close to the anlage of the pineal gland (dorsal diencephalon) and the dorsal mesencephalon. This region was mapped previously in uncaging experiments in zebrafish to contribute to the eye [50] and corresponds to the rostral limit of NC formation mapped in the newt *Pleurodeles* [51], chicken embryos [1], mouse [52] and human [53]. In mice, a group of periocular NC cells with a reduced neuropilin-1 (Npn-1) expression emerges from Npn-1 expressing NCs to migrate over the lens, i.e. into the distal end of the eye, and establish the corneal endothelium [54]. The 2°NC cell population is possibly related to this mouse NC population.

   Both 1°NC and 2°NC cells share *sox10* gene expression. However they differ in the maintenance of sox10 expression. *sox10* mRNA expression is not detectable in NC cells once they have entered the eye. *sox10*:*mem-tdEosFP* transgene expression had ceased in 1° and 2°NC cells with their immigration into the eye at 13–14 hpf and 22–28 hpf respectively, However, EosFP fluorescent protein still persisted until around 48 hpf leaving a memory of former *sox10* driven gene expression in the NC cell derivatives.

   There are two peaks of Wnt reporter expression in ocular NC cells. The early peak (12–14 hpf) includes the precursors of 1°NC cells as previously described [21]. The second peak of Wnt reporter activity from 18 hpf onwards was noted in 2°NC cells. Prior to migration into the eye, half of *sox10*:*mem-tdEosFP*-positive 2°NC cells co-express the Wnt reporter. They continued to express the Wnt reporter after 2°NC cells entered the nascent anterior segment. The role of Wnt signalling in 2°NC cells remains to be determined. Blocking Wnt signalling with the Wnt signalling inhibitor IWR-1-endo led to a 50% reduction of premigratory *sox10*-positive 2°NC cells suggesting a role of Wnt signalling in the control of the number of premigratory 2°NC cells.

Both NC cell groups show distinct transformations of cell behaviour: 1˚NC cells retain cell-cell contact over large cellular areas while enwrapping the back side of the eye and display a cuboidal cell shape. The 2˚NC cells initially accumulate as a group of cells at the dorsal rim of the eye before entering the eye as individually migrating cells. This switch of 2˚NC cells from clustering to migration as individual mesenchymal cells is correlated with their entry into the eye and requires *pax6a/b* as 2˚NC cells failed to enter the eye in *pax6a/b* morphants/mutants. Similarly, aggregation of NC cells was observed at the dorsal edge of the nascent eye in rat Pax6-/- embryos [55]. Pax6a and Pax6b are not significantly expressed in 2˚NC cells at this stage, indicating that Pax6a/b expression in the surrounding tissues is responsible for guidance, as previously also suggested for the abnormal migration of NC cells in Pax6 heterozygous mutant mice [7] and Pax6 homozygous mutant rats [55]. In agreement with these findings, we found an altered expression of cell guidance molecules in ocular and surrounding tissues in the zebrafish *pax6a/b* mutants.

We have focused on the development of the corneal endothelium as a well-defined NC-derived tissue of the eye. By 26–28 hpf, around 20 mesenchymal-shaped 2˚NC cells have reached the nascent AS. Corneal endothelium formation and reporter expression commences only by 60 to 70 hpf, as judged by expression of the *Tg(corneaEndo:GFP)*, with the first ten cells expressing the transgene in the corneal area juxtaposed to the lens. These cells then proliferate until on average ca. 27 cells cover the entire inner surface of the cornea by 120 hpf. Curiously, endothelial cells form concentric rings, which may be a reflection of their patterns of division and migration over the inner surface of the cornea.

## The role of the lens and TGFβ signalling in shaping the anterior segment

Ectopically located 2˚NC cells in *pax6a/b* mutants/morphants maintain high levels of *sox10* transgene expression, suggesting that these cells have to enter the eye for turning off *sox10* expression. This behaviour indicates that *sox10* is not directly required for endothelial differentiation. However, when we examined the *sox10^{t3/t3}* mutant *colourless*, the endothelium was not properly formed in this mutant, suggesting that *sox10* is involved in corneal endothelial formation. Interference with TGFβ signalling prevents differentiation of the corneal endothelium. The requirement of TGFβ signalling is reminiscent of the effect of TGFβ signals in *in-vitro* cultures of mouse NC cells [48]. The zebrafish lens strongly expressed *tgfb2* between 22 and 40 hpf, covering the time of immigration of NC cells into the AS and their differentiation into the various AS structures including the corneal endothelium. In addition expression was noted in mesenchymal cells in both the proximal and distal compartment of the eye. This was paralleled by the expression of the sensor of TGFβ signalling *Tg(smad3:gfp)*. In this context, it may be of importance that the aqueous humour of the anterior chamber is a source of TGFβ molecules in the mammalian eye [56]. TGFβ molecules may thus be freely diffusible throughout the anterior chamber. Pax6a/b compound mutants lack the lens entirely and may thus influence anterior chamber formation by indirectly eliminating lens derived TGFβ. Our TGFβ inhibitor experiments support a necessary role of TGFβ in corneal endothelium development.

There is evidence of direct regulation of TGFβ2 by Pax6 from studies in mammals. In mouse, Pax6 binding sites were identified in the *Tgfb2* promoter [57], and a ChIP-seq/transcriptome study reported up-regulated *Tgfb2* transcripts in conditional Pax6 homozygous mutant lenses during embryonic stages [58,59], suggesting that Pax6 negatively regulates *Tgfb2* transcription. We found that, when 2˚NC cells reached the eye lenticular *pax6b* expression is polarized to the distal side, while *tgfb2* and *smad3:gfp* reporter transcripts are found on the opposite, proximal side of the lens. This is in agreement with the hypothesis that the distal *tgfb2* and *smad3* reporter expression were repressed by *pax6a/b*. In contrast, a mouse study

reported down-regulated *Tgfb2* transcripts in Pax6(+/-) heterozygous mutant lenses at postnatal day 1 when the AS is already established [57]. Thus, whether Pax6 activates or represses its target might be controlled by stage specific co-regulators in the lens.

Removal of the lens from wild type eyes at 24 hpf phenocopied the defects in the annular vasculature seen in *pax6a/b* mutants at 5 dpf. When a wildtype lens was transplanted into a *pax6a/b* double mutant eye, the vasculature formed normally in the mutant, showing that a wildtype lens can direct vessel formation in a mutant background. Whether these effects are mediated by lens derived TGFβ signals remains to be seen. The trabecular meshwork is a key NC-derived anterior segment structure for intraocular pressure control, together with lymph-derived Schlemm's canal. Although the homologue of the mammalian trabecular meshwork has not been described in zebrafish, the aqueous humour flows out through a ventral canalicular network considered to be of ocular mesenchymal origin [14]. Pax6a/b mutants appear to have defects in the blood/aqueous humour barrier, presumably linked to a failure to establish the annular vasculature correctly.

## Pax6 mutations cause ASD in zebrafish without affecting the overall polarity of the eye

In mammalian studies of Pax6 function, it has been proposed that *pax6* establishes distal identities such as the lens, the iris and the cornea in a dose-dependent manner within the eye [9,11]. In addition, Pax6 is recently shown to be required for TGFb2 expression in the chick optic vesicle to establish the eye's first axis [60] and in the mouse iris/ciliary body [61]. The duplication of the genome at the base of teleost evolution has generated two *pax6* alleles that have been retained with overlapping functions in the zebrafish genome, offering a unique opportunity to study the influence of *pax6* gene number on eye development. In agreement with mouse studies, we noted a tendency of increasing phenotypic penetrance and severity of the anterior chamber defects with decreasing numbers of wild type *pax6a/b* copies in the zebrafish.

In concordance with mouse homozygous *pax6* mutant phenotypes, zebrafish *pax6a-/-; pax6b-/-* double homozygous mutants showed severe defects in distal eye structures, lacking the lens, iris, corneal endothelium and ganglion cell layer of the retina. With the exception of the missing ganglion cell layer, the proximal side of the eye was relatively unaffected, featuring normal retinal pigment epithelium, photoreceptor layers, outer nuclear layer, outer plexiform layer, inner nuclear layer and inner plexiform layer. In contrast to mouse *pax6* homozygous mutants [62], zebrafish compound mutants also formed nasal pits.

The milder phenotype in zebrafish may be the result of the specific mutant alleles. Missense mutations in human Pax6 lead to a variety of phenotypes ranging from mild to severe AS defects [63]. The *pax6b* mutant allele *sunrise* used in this study carries a missense mutation L244P in the homeobox DNA-binding domain. The penetrance of its mutant phenotype (smaller lens with elliptic shape and coloboma) was shown to be modulated by the chaperon Hsp90 [64]. In contrast to the *pax6b/sunrise* allele used here, the *pax6a* mutant allele harbours a premature termination codon, lacking the homeobox DNA binding domain and the PST-rich transactivation domain. Importantly, the phenotypic expression and, thus, the interpretation of the data with respect to subfunctionalization as well as gene dosage may also be hampered in addition by the presence of maternal *pax6a/b* mRNAs.

1°NC cells, which normally settle only in the proximal part of the eye, aberrantly invade distal locations in *pax6a/b* double mutant eyes. During early mammalian eye development, Pax6 appears to be a critical determinant of polarity along the proximodistal axis of the eye. Accordingly, the highest Pax6 levels and dependency are found in the distal part of the eye anlage

[10,11]. As a consequence of Pax6 mutation, proximal markers expanded to the distal site of the mouse eye [12]. When we examined the zebrafish *pax6a/b* double mutants, we did not find consistent evidence for a disturbed patterning of the eye along the proximodistal axis. The reduction in lens specific markers may reflect a requirement for *pax6* expression during lens development. Given the similarity in the phenotypes of the *Pax6* mouse and zebrafish mutants, we uncovered a later layer of regulation, which had most likely been missed in mammals due to the more severe phenotype. It is tempting to speculate that, through the fortuitous choice of the *pax6b/sunrise* allele and the presence of maternal contributions of *pax6a* and *pax6b* in the zebrafish, the residual *pax6* activity may have overcome the early patterning defects, thereby revealing the later direct function of *pax6* in the development of the anterior eye segment.

## Pax6a/b are required for expression of cell guidance molecules

NC cells are misrouted with high penetrance in *pax6a/b* double mutants: 1˚NC colonise the entire eye instead of colonising the proximal half of the eye only as in wild type embryos. The 2˚NC cells, which normally migrate into the nascent AS, are excluded from the eye in *pax6a/b* double mutants. Abnormal migration of NC cells was also observed in Pax6 heterozygous mouse mutants [7]. Pax6 was not detectably expressed in premigratory and migratory NC cells. Thus, misrouting of NC cell migration appears to be caused by non-cell autonomous factors. In agreement with these findings, we identified changes in the expression of cell guidance molecules in the eye and the surrounding tissue. Some guidance molecules such as *efnb2a* were affected only strongly in *pax6a/b* double mutants. Other guidance molecules such as *efna3b* and *ephb3a* also showed a reduction in expression in *pax6b* single mutants. These findings suggest that expression of these guidance molecules is sensitive to the number of wild type *pax6* copies. This dependence of the expression of cell guidance molecules on Pax6 is reminiscent of the role of Pax6 in the expression of axon guidance molecules controlling proper axonal connections in the forebrain [65]. In the embryonic lens of mice, Pax6 directly regulates Semaphorin ligands including s*ema3aa* [58,59] that prevent *Nrp1*-expressing NC cells from migrating over the lens during cornea development [54]. s*ema3aa* expression is absent in *pax6a/b* double mutants. Abnormal migration of 1˚NC cells in double homozygous *pax6a-/-;pax6b-/-* mutants may therefore be related to the lack of the lens and, therefore, early s*ema3aa* expression, which normally prevents 1˚NC cells from entering the distal eye. However, s*ema3aa* cannot be the only factor: In a significant proportion of *pax6b-/-* embryos, ocular NC cells were also misrouted even though *sema3aa* was expressed. Moreover, knockdown of *sema3aa* translation did not show an effect on NC migration in the eye, suggesting either redundancy with other guidance cues or no function. Knocking-down *nrp2b* mRNA translation, however, lead to an eye phenotype resembling that of *pax6a/b* mutants. Although we did not observe an effect when we knocked-down *sema3aa*, the requirement of the semaphorin co-receptor neuropilin *nrp2b* indicates a role of Semaphorin signalling in the guidance of the NC cells into the eye.

Interestingly, 1˚ and 2˚NC show differential requirement of *nrpb2* function. Migration of 2˚NC cells was not affected by *nrpb2* knock-down suggesting distinct guidance mechanisms in the two populations. 2˚NC cells enter through the dorsal side of the eye. In this context, it may be of relevance that the dorsal expression of *efnb2a* and *efna3b* is attenuated in *pax6b* single mutants and abolished in *pax6a/b* double mutants. In both genetic backgrounds, the migration of 2˚NC cells is affected. Clearly, expression of these markers shows a dependence on the *pax6a/b* gene dose.

In conclusion, the severe ASD phenotype with lacking lens, corneal endothelium and vasculature in *pax6a/b* homozygous double mutants is tightly linked to the resultant misguidance of the two distinct ocular NC populations. We showed that Pax6 promotes guidance molecule expression in the ocular cup and its surrounding mesenchymal cells. Later defects in the anterior segment of the eye are attributable at least partially to the lack of the lens as an important source of further signals such as TGFβ. These findings may have implications for glaucoma development in congenital cataract patients whose lens are typically removed in the first few weeks after birth.

We have characterised in depth the development of the AS of the zebrafish eye and showed how these processes are disturbed in a zebrafish model for human ASD. The knowledge on NC function in eye development gained with this model may contribute to future therapeutic approaches. Given the ease with which zebrafish embryos can be exposed to drug candidates, our detailed description of zebrafish AS development and the new genetic and transgenic tools will pave the way to drug development. By exploiting the advantages of a whole organism approach offered by the zebrafish embryo, screening of this system will probably identify new chemical entities and drug targets representing previously unrecognised drugable components in the complex developmental pathways disturbed in ASD and congenital glaucoma.

## Methods

### Ethics statement

Animal husbandry and experimental procedures were performed in accordance with the German animal protection regulations and were approved by the Government of Baden-Württemberg, Regierungspräsidium Karlsruhe, Germany (AZ35-9185.81/G-137/10, AZ35-9185.81/G-22/15, AZ35-9185.81/G-102/19 and AZ35-9185.81/G-103/19).

### Fish, transgenic lines

We used the wildtype line ABO (European Zebrafish Resource Centre [EZRC], Karlsruhe). Zebrafish (*Danio rerio*) embryos were maintained at 28˚C as described [66,67]. The 7.2 kb *sox10* regulatory element [19] was used to drive expression of Eos fluorescent protein (tandem dimer tdEosFP [18]). *Tg(sox10:mem-tdEosFP)* [official ZFIN name *Tg(-7.2sox10:gap43-tdEosFP)*$^{ka97}$] express tdEosFP N-terminally tagged with a zebrafish *gap43*-derived dual palmitoylation sequence for plasma membrane localization. *Tg(sox10:h2a-tdEosFP)* [official ZFIN name *Tg(-7.2sox10:h2afva-tdEosFP)*$^{ka96}$] harbour tdEosFP fused N-terminally with *h2afva* for nuclear localization. Transgenic lines were generated by injection of Tol2kit-based DNA constructs [68] into one-cell stage embryos, followed by screening for positive founder fish in the next generation. The Wnt reporter line *Tg(7xTCF-Xla.Siam:nlsmCherry)* [22] was crossed with each Sox10 reporter line and double-positive embryos were used for imaging. Homozygous mutant *pax6b*$^{tq253a/tq253a}$ zebrafish were viable (the *tq253a* allele is also called *sunrise*), fertile and maintained as adult fish. *pax6a* mutants were created by deleting a ~3 kb genomic region across exon 8–12 with CRISPR/Cas9, as described in S11 Fig. Briefly, two short guide RNAs (sgRNAs) were transcribed *in vitro* by using the MEGAshortscript T7 transcription kit (Ambion) from double stranded DNA templates prepared as described [69], harbouring target sequences of 5'-GGT TGC CAA CAG TCA GAC GG-3' for *pax6a* exon 8 and 5'-GGT CTG GCT GGG CTG TGA AG-3' for exon 12. Two sgRNAs, each at 300 ng/µl final concentration, were incubated briefly with GeneArt Platinum Cas9 nuclease (300 ng/µl final; Invitrogen) and injected into the zebrafish one-cell stage blastodisc.

## Genotyping

Genomic DNA was prepared by an adopted HotSHOT method [70], using a base solution (25 mM KOH, 0.2 mM EDTA pH13.8) and a neutralisation solution (40 mM Tris-HCl, pH3.8). A part of the tail fin or the whole embryo was incubated in 75 μl of base solution at 96˚C for 20 min and cooled down to room temperature. 75 μl of neutralisation solution were added to the sample before KASP assay or genotyping PCR. KASP assay was performed to genotype *pax6b* mutants harbouring a single base mutation in the *pax6b* locus. The following three primers, two-allele specific forward primers (wild type and mutant alleles) and a common reverse primer, were synthesized by LGC Limited (UK): (wild type allele labelled with FAM: 5'-AAA GTT GTG ATC GTT CAC CTT TCT CAA-3'; mutant allele labelled with HEX 5'-AGT TGT GAT CGT TCA CCT TTC TCA g-3' [mutation in small letter]; common primer: 5'-CGT CCT TCA CGC AGG AGC AGA T-3'). KASP assay mix (0.11 μl), KASP master mix (4 μl), 1 M MgCl$_2$ (0.64 μl) and genomic DNA (1 μl) were assembled per reaction in a 96-well plate (AB-0800/K, Thermo Fisher Scientific). After PCR, fluorescence was measured by an EnVision plate reader (PerkinElmer) and analysed by KlusterCaller software (LGC Limited, UK). To genotype *pax6a* mutants, we designed two sets of primers, one for detecting the deletion and another for the presence of the wild type allele. A set of pax6a_Ex8-12_L (5'-GTG GCC GTA GCC TAA TTG AA-3') and pax6a_Ex8-12_R (5'-GCC ATT GAT TGG CTG TTC AT-3') was designed to yield a 405 bp amplicon for identifying the mutant allele, and another set pax6-a_Ex8_L (5'-GTG GCC GTA GCC TAA TTG AA-3') and pax6a_Ex8_R (5'-AGT GCG ATT CCT TTG CAG TT-3') was targeted to yield a 384 bp amplicon for identifying the presence of the wild type allele.

## Morpholinos

Following antisense morpholino oligos were synthesized by Gene Tools, LLC (OR, USA) and injected into one-cell stage embryos at concentrations indicated in parenthesis: 1ATG_pax6ab (250 μM): 5'-GTTATGGTATTCTTTTTGAGGCATT-3'; 2ATG_pax6ab (250 μM): 5'-ACTGTGACTGTTTTGCATCATGGAC-3'; 1ATG_pax6ab_5-mismatch (250 μM): 5'-GTTAaGcTATTCTTaTTcAcGCATT-3' (small letters indicate the mutated sites in mismatch morpholinos);

2ATG_pax6ab_5-mismatch (250 μM): 5'- ACTcTcACTcTTTTcCATCATcGAC-3'; MO2-nrp2b (125 μM) [40]: 5'- CGCGTAGAGGAAAAAGCTGAAGTTC-3'; MO1-sox10 (250 μM) [31]: 5'- ATGCTGTGCTCCTCCGCCGACATCG-3'; MO4-tp53 (62.5 μM) [71]: 5'-GCGCCATTGCTTTGCAAGAATTG-3'

## *In situ* hybridization

Whole-mount *in situ* hybridisation was conducted as described [72]. Antisense digoxigenin-labelled RNA probes for *otx5* [73], *wnt2* [74], *sox10*, *pitx2*, *pax6b*, *foxc1a*, *zgc:92380* [16], *sox9b*, *tgfb2*, *gfp*, *efna3b*, *epha4b*, *aldh1a2*, *efnb2a*, *vax2*, and *sema3aa* were prepared from cloned DNA constructs. For double fluorescent *in situ* hybridisation, a set of two probes, each labelled either with digoxigenin-11-UTP (Roche) or DNP-11-UTP (Perkin Elmer), was used during hybridisation (day 1). After stringency washing and blocking in 1% BSA/PBS/0.1% Tween-20, embryos were incubated with a 1:1000 dilution of anti-digoxigenin-POD (poly) Fab fragments (Roche; day 2) and labelled with a 1:250 dilution of TSA Plus Cy3 stock solution (Perkin Elmer) for 45 min, then further incubated in PBS/0.1% Tween-20 (PBT) at 4˚C overnight (day 3). After a PBT wash, embryos were treated with 3% H$_2$O$_2$ for 2 h to eliminate the activity of probe-bound peroxidase. Embryos were extensively washed in PBT and incubated with a 1:500 dilution of anti-DNP-HRP (Perkin Elmer) at 4˚C overnight (day 4). The second

fluorophore labelling was done with a 1:250 dilution of TSA plus Cy5 (Perkin Elmer; day 5). For nucleus counterstaining, we applied incubation on a 1:1000 dilution of DAPI in PBS/0.5% Triton-X100 at 4˚C overnight (day 6).

## Histology

Epoxy resin (EPON) embedding and sectioning were done as described [16]. Briefly, the embryos were dehydrated through a series of increasing concentrations of ethanol (50%, 70%, 95%, 100% [v/v]), followed by 100% propylene oxide and a series of increasing concentrations (33%, 66%, 100%) of EPON (glycid ether 100, Serva) in propylene oxide. Polymerisation was carried out at 65˚C in the presence of 20.8% (w/w) DDSA (dodecenylsuccinic acid anhydride, Serva) and 23.3% (w/w) MNA (methylenacid anhydride, Serva), with the accelerator 1.8% (w/w) DMP 30 (2,4,6-tris[dimethyl-aminomethyl]phenol, Serva). The polymerised block was trimmed (Leica EM TRIM) and 6 μm and 350 nm sections were cut with glass and diamond knives (DiATOME ultra 45˚) using RM2065 and UC7 microtomes (Leica Microsystems), respectively.

## Intravenous dye injection

Zebrafish embryos at 5 dpf were anesthetized in 643 μM MS-222 (tricaine methanesulfonate) and embedded in 0.8% low melting point agarose laterally with the right side up. To minimize the effect of low blood pressure on the aqueous humour dynamics, embryos were immersed in fish water to wash out the anaesthetic after hardening of the agarose. A mixture of 2000 kDa FITC-labelled dextran and rhodamine-labelled 3 kDa dextran was transferred into a glass capillary and 10–20 nl was injected into the cardinal vein in the vicinity of the heart by a microinjector (Eppendorf FemtoJet) as described [75].

## Multidimensional real-time imaging

Measurements were performed on a custom-built digital scanned laser light sheet fluorescence microscope (DSLM) setup that allows multi-channel image acquisition at high temporal and space resolution assisted by bi-directional sample illumination [23]. The spectrum for illumination included lasers of 405 nm (diode laser module, Omicron LuxX, 60 mW), 488 nm (diode laser module, Omicron LuxX, 60 mW) and 561 nm (DPSS laser, Cobolt Jive, 75 mW). With the 5×/0.15 (illumination) and 16×/0.8w (detection) objective lenses (both Nikon, Japan), the lateral and axial full width at half maximum of the point spread function resulted in 1.5 ± 0.3 μm and 6.6 ± 1.4 μm at 0.4 μm and 2.0 μm voxel size in the lateral and axial directions, respectively. Embryos were mounted in cleared fluorinated ethylene propylene tubes in 0.3% (w/v) low-melting point agarose. The pipeline of algorithms for NC cell tracking has been described earlier [25].

Confocal microscopy images were acquired with a Leica TCS SP5 upright microscope with an HCX PL APO 20x/0.70 lambda blue IMM CORR objective. Time lapse imaging was conducted in resonant scanning mode (8 kHz, bidirectional scanning). For photoconversion, filter cube A (excitation bandpass filter, 360 ± 40 nm) was selected and the sample was illuminated for 20 s. The temperature was set to 28˚C. Embryos were mounted in 0.5% (w/v) low-melting point agarose in embryo medium (50 mM NaCl, 0.17 mM KCl, 0.33 mM $CaCl_2$, 0.33 mM $MgSO_4$, 0.7 mM HEPES pH7.0). Co-localization of fluorescent signals was analysed in ImageJ "Image Calculator" by using the "Multiply" operation between two channels after background subtraction.

## Inhibitors

SB-505124 (Sigma-Aldrich), an inhibitor of TGFβ type I activin receptor-like kinase, was used at 30 μM. IWR-1-endo was obtained from Dr. Lawrence Lum. As a negative control, embryos were treated with 0.5% (v/v) DMSO in embryo medium.

## Statistical analyses

Statistical significance was assessed by using R [76]. All numerical data underlying graphs and statistics are summarized in S2 Method.

## Supporting information

**S1 Video. Time-lapse movie of a *Tg(sox10:h2a-tdEosFP)* embryo during 10–33 hpf.** The maximum intensity projection of 3D image stacks obtained with DSL microscopy is shown. Dorsal view of the embryo is shown with the anterior end oriented toward the top. Time-lapse interval is 2 min. The fluorescence intensity of the nuclei of NC cells is shown in blue-green-yellow colours from lower to higher intensities.
(M4V)

**S2 Video. Confocal time-lapse of *Tg(sox10:mem-tdEosFP)* transgenic embryo during 15–24 hpf.** In the first half of the movie (up to a stop at 18 hpf), 1˚NC cells migrate over the proximal side of the optic cup, sliding over the back surface of the eye. In the second half of the movie, note the cellular motility of individual 2˚NC cells during 18–22 hpf, constantly exchanging neighbours within the cluster. Transformation of 2˚NC cell behaviour is evident after 22 hpf, when the cell clusters dissolve at the dorsal edge of the eye and migration into the distal side of the eye begins as individual cells.
(M4V)

**S3 Video. Wnt reporter activation in NC cells.** Time lapse analysis of a *Tg(sox10:h2a-tdEosFP;Wnt-rep)* double transgenic embryo imaged by DSLM (the same embryo shown in Fig 2A–2E). Colocalized signals of h2a-tdEosFP and Wnt reporter (white) is merged over the h2a-tdEosFP fluorescence (blue) in NC cells (dorsal view; anterior left). Two waves of Wnt activation were observed in NC cells.
(MP4)

**S4 Video. Overlay of cell trajectories on raw microscopic images.** (Left half of the movie) Trajectories of a 2˚NC lineage are shown with a temporal colour code, superimposed over the local maximum projection images of a *Tg(sox10:mem-tdEosFP)* embryo acquired by DSLM imaging. A dorsal view of the left eye is shown with rostral toward the right. Green spots are centres of the nuclei. Every four time points of the original time sequence acquired at 26 s interval is shown to decrease the file size. Arrows indicate the position of the tracked nucleus and its daughter cell nuclei. An arrow with asterisk indicates the cell shown on the right side of the movie. Note that all daughter cells in this example migrated into the distal side of the eye, adopting the fate of 2˚NC. The location of the optic cup is given at the beginning and the end of the movie. (Right half of the movie) A cell-centred view of the same movie is shown, focusing on a selected 2˚NC daughter cell.
(M4V)

**S5 Video. Trajectory path comparison of 1˚NC and 2˚NC cells during 11–24 hpf.** DSLM time-lapse data from a *Tg(sox10:mem-tdEosFP)* embryo were processed to track individual NC cells. Trajectories from 1˚NC and 2˚NC cells are shown at the bottom and top,

respectively. The second half of the movie shows the cumulative path of NC migration during 11–24 hpf from various viewing angles.
(MP4)

**S6 Video. Locally labelled 2˚NC cells migrate into the distal compartment of the eye.** DSLM time-lapse movie from a *Tg(sox10:mem-tdEosFP)* embryo during 18–30 hpf. Dorsal view with the anterior (rostral) side up. Two clusters of 2˚NC cells between the two eye primordia were illuminated by a short pulse of 405 nm laser light to convert the green fluorescence emitting form of mEos into the red emitting form. Arrows indicate locally labelled 2˚NC cells. *De novo* expression of green mEos reporter in the photoconverted cells makes them appear in white.
(M4V)

**S7 Video. Histological reconstitution of the *pax6a-/-;pax6b-/-* eye at 5 dpf.** Semi-thin 350 nm-thick sections of a *pax6a-/-;pax6b-/-* embryo were cut, individually collected on a glass slide, counter stained with Toluidine blue-O and mounted in EPON. Images were acquired with a 20x/0.5 objective and manually aligned to reconstitute a three-dimensional overview of the eye using the TrackEM2 plugin in ImageJ/Fiji. Note the normal proximal structures in the eye, including the retinal pigment epithelium (RPE), photoreceptor/outer nuclear layer (ONL), outer plexiform layer, inner nuclear layer (INL), and inner plexiform layer (IPL). Both the optic nerve and the optic artery that runs in parallel were formed normally. In contrast, the distal side of the eye is severely affected with no formation of the lens, the anterior chamber and the ganglion cell layer (GCL) of the retina. Scale bar: 50 μm.
(M4V)

**S8 Video. *pax6a/b* are required for normal corneal endothelium formation.** Time lapse analysis of the eyes of *Tg(corneaEndo:GFP)* embryos with wild type *pax6a/b* loci (left, *n* = 5 embryos), homozygous mutant *pax6b-/-* (middle, *n* = 3 embryos), and *pax6a*-gRNA-injected *pax6b-/-* (right, *n* = 3 embryos) during 72–89 hpf. Images were acquired every 10 min. Maximum projection of tangential views of the eye is shown. Reduction of Pax6 dose correlates with the defect in the corneal endothelium formation. Scale bar: 50 μm.
(M4V)

**S9 Video. Pax6a/b are required for guiding the two ocular NC populations to their distinct destinations.** Photoconversion of *Tg(sox10:mem-tdEosFP)* embryos at 17 hpf visualizes 1˚NCs and 2˚NCs separately in magenta and green, respectively. Lateral views of time lapse fluorescent images were merged over the bright field images. The same embryos shown in Fig 7 are presented as movie. Note that, in the *pax6a/b* double homozygous embryo the, 2˚NC cluster persisted at the dorsal edge of the eye.
(M4V)

**S10 Video. TGFβ signalling inhibitor blocks the formation of the corneal endothelium.** *Tg (corneaEndo:GFP)* embryos were treated with either 0.5% DMSO (upper row) or 30 μM SB-505124 (lower row) during 46–87 hpf and en face views of the eye were acquired by confocal imaging every 11 min. Four representative biological repeats are shown for each chemical treatment. Green fluorescence is shown in a pseudo colour code as indicated on the left. Corneal endothelial cells appear in green/yellow. Scale bar: 100 μm.
(M4V)

**S1 Fig. 1˚NC cells cover the proximal but not the distal side of the optic cup.** (A-C') 3D rendered images of a *Tg(sox10:mem-tdEosFP)* embryo are shown from three different angles: dorsal views (A-A'), lateral views (B-B') and ventral views (C-C'). Anterior, left. Two selected time

points (A-C: 15 hpf; A'-C': 18 hpf) are shown. The grey spheroid represents the optic cup (OC). One side of the eye (rectangular region) in the dorsal view (A-A') is highlighted in lateral (B-B') and ventral views (C-C'). (A-C) At 15 hpf, 1˚NC cells start covering the optic cup from its posterior end and its proximal side (arrowheads, A). Only a small dorsal portion of the eye is covered by 1˚NC cells (arrowheads in B, arrow in C), while the ventral half of the eye is not covered with 1˚NC cells (C). The arrow in C points at 1˚NC cells on the dorsal side of the eye. (A'-C') By 18 hpf, 1˚NC cells have reached the rostral end of the optic cup (arrowhead in A'), with the proximal side of the eye covered roughly halfway (arrows in B' and arrowheads in C'). Importantly, 1˚NC cells exclusively migrate over the proximal side of the eye (arrows, B') and do not migrate over the lens side (distal side; arrowheads, B'). Note that 2˚NC cells in clusters (asterisks in A') have not yet reached the eye at 18 hpf. (D-F) Ultrastructure of 1˚NC cells (magenta). (D) 1˚NC cells are located on the proximal side of the optic cup (blue; transverse view at 18 hpf). The regions indicated by squares are shown in E and F. Black honey comb pattern is from the grid. (E) 1˚NC cells form a mono-layered sheet. (F) A 1˚NC cell (coloured in magenta) show a large cell-cell contact plane with neighbouring 1˚NC cells. OSE: optic surface epithelium; Br: brain.
(PDF)

**S2 Fig. 2˚NC cells turn off _sox10_ expression upon migration into the eye.** (A-C) _in situ_ gene expression analysis of _sox10_ (A), _pitx2_ (B) and _foxc1a_ (C) in WT embryos at 28 hpf. Arrows indicate peri-ocular mesenchymal cells. Scale bar: 50 µm. (D-H') Photoconversion of mem-tdEosFP and recovery of green fluorescence for visualization of _de novo sox10_ reporter expression. At the indicated stages, mem-tdEosFP was switched from green to red (magenta) by illumination with 400-nm light (D-H). Continued mem-tdEosFP expression was scored by reappearance of green fluorescence at 36 hpf (D'-H'). Recovery of green mem-tdEosFP fluorescence in the eye at 36 hpf was noticed in embryos photoconverted until 20 hpf (D-F'). Photoconversion at 22 or 24 hpf showed recovery of green fluorescence in the head but not in the eye region (G'-H'), indicating that 2˚NC cell derivatives which have migrated into the eye have turned off _sox10_ transgene expression by 22 hpf. In contrast, _sox10_ reporter expression in NC derivatives has ceased in the pharyngeal arches at 16 hpf (asterisks, A') as well as in 1˚NC on the proximal side of the retina. Green NC cells in panel E (white arrow) are un-photoconverted cells. Scale bar: 40 µm.
(PDF)

**S3 Fig. Wnt signalling is required for the correct number of 2˚NC cells but is dispensable for formation of the corneal endothelium.** (A): Embryos were treated with either 1% DMSO alone or the Wnt inhibitor (10 µM IWR-1, 1% DMSO) from 6 to 22 hpf or 18 to 22 hpf. The phase of active _sox10_ expression in 2˚NC cells is indicated. (B-D): _sox10:mem-tdEosFP_ (green) and Wnt-rep (magenta) expressing embryos were exposed to solvent (B) or Wnt inhibitor (C) from 6 to 22 hpf and fixed and counter-stained with DAPI (blue). Inhibition of canonical Wnt signalling abolished the expression of the Wnt-rep (magenta, arrowheads, B) and reduced the expression of mem-tdEosFP (C). Quantification of endogenous _sox10_ mRNA positive cells (D) reveals a significant decrease of _sox10_ expressing cells anterior to the otic vesicle at 22 hpf for both exposure windows ($p < 0.001$, $n = 24$ embryos, 6–22 hpf, $n = 21$ embryos, 18–22 hpf) compared to controls ($n = 26$). (E-H): Ultrastructure of the corneal endothelium at 5 dpf in control (E, G; $n = 3$ embryos) and IWR-1 treated embryos (6–22 hpf, F, H; $n = 3$ embryos). Transverse sections through the centre of the lens. Inhibition of Wnt signalling caused formation of an abnormal corneal epithelium (Ep) with the inner and outer epithelial layers being separated by a large oedema. No effect on the formation of the corneal endothelium (En) or the thickness of the stroma (St) was notedic (G, H; $n = 3$). Orientation of embryos: B-C:

anterior up, view onto dorsal. Scale bar: B-C: 70 μm; E-F: 6 μm; G-H: 2 μm (I) Dorsal view of the eye from *Tg(sox10:h2a-tdEosFP;Wnt-rep)* line at 28 hpf. Anterior left. Approximately 50% of NC cells (23/47 cells) in the distal side of the eye (green, left panel) express Wnt reporter (right panel). le: lens.
(PDF)

**S4 Fig. Backward and forward tracking from a second embryo.** Tracking analysis of 1˚NC and 2˚NC cells from a second embryo double transgenic for *Tg(sox10:h2a-tdEosFP)* and *Tg (Wnt-rep)* that was imaged essentially under the same conditions as the one shown in Fig 2. Two groups of h2a-tdEosFP-positive NC cells were selected at 17 hpf for systematic tracking as shown in Fig 2; 1˚NC cells that were already in contact with the optic cup (112 cells; yellow circles in B, B', D, D') and 2˚NC cells that formed the temporal cell cluster next to the diencephalon and mesencephalon (62 cells; yellow circles in C, C', E, E'). B-E, dorsal projection with rostral left; B'-E', lateral projection with dorsal up. Tracks are shown with a temporal colour code shown in the panel A. 2˚NC cells at the dorsal edge of the eye originated from the diencephalon and mesencephalon (asterisk, C) and migrated into the anterior chamber of the eye from the dorsal side of the retina (arrow in E'). Scale bar: 100 μm.
(PDF)

**S5 Fig. Early origin of ocular NC cells.** (A-B) Backward tracking analysis of 1˚NC and 2˚NC cells in a *Tg(sox10:h2a-tdEosFP)* embryo at 14 hpf. At 17 hpf, H2a-tdEosFP-positive cells that were already in contact with the optic cup were selected for 1˚NC tracking (*n* = 112 cells) and those found at the diencephalic and mesencephalic regions for 2˚NC tracking (*n* = 62 cells). Positions of 1˚NC (A) and 2˚NC cells (B) at 14 hpf are shown in green, merged over the dorsal maximum projection view. Red dots are the centres of the nuclei. Anterior is left. (A) 1˚NC cell nuclei (green spots, 25 out of the selected 112 cells) are distributed in the anterior part of the embryo, in relatively narrow lateral regions along the anterior-posterior axis (less than 10-cell diameters, horizontal double-headed arrow). Along the mediolateral axis, 1˚NC cells are found laterally within the lateral NC stripes (stippled line enclosed area). Within each lateral NC stripe, 1˚NC cells show no obvious mediolateral bias. (B) In comparison, 2˚NC cell nuclei (6 green spots in B) are found biased toward the medial part of the embryo within the narrow anteroposterior region in which 1˚NC cells are found (horizontal double-headed arrow).
(PDF)

**S6 Fig. 2˚NC cells show distal-oriented movement after dissolving the cluster.** (A, B) Movement angle histogram for NC cells destined for proximal (A, 1˚NC cells; *n* = 44) and distal (B, 2˚NC cells; *n* = 37) sides of the eye. Movement trajectories of an individual NC cell were projected on a horizontal plane and the angle relative to the anteroposterior axis was measured every 10 time points (every 3 min 24 s). Each bin is 10˚. Black arrows show median direction (56.5˚ and 83.6˚ for proximal and distal selection, respectively). Wallraff test of angular distances between two groups showed significant difference, *p*-value $< 2.2 \times 10^{-16}$. (C, D) Centred tracks starting from 22 hpf for proximal (C, 1˚NC; *n* = 44) and distal (D, 2˚NC; *n* = 37) destined NC cells. The starting position is set to 0 (bottom centre). (C) NC cells on the proximal side of the eye show moderate distal-oriented movement after 22 hpf. (D) In contrast NC cells destined for the distal side of the eye show higher displacement toward the distal end. (E-K) Cell morphology of 1˚NC and 2˚NC cells. (E-F) A 18-hpf embryo from *Tg(sox10:mem-tdEosFP)*, highlighting NC cells in green. (E) 3D-overview of the optic cup region (stippled circle). (F) Projected focal planes for the proximal side of the eye (stippled region). Note the cuboidal or hexagonal cell shape of 1˚NC cells retaining large areas of cell-cell contact with

neighbouring cells. (G-K) Overview of a 26-hpf *Tg(sox10:mem-tdEosFP)* embryo (G, maximum projection) and magnified single cell morphology of NC cells in a single focal plane for proximal (H-I) and distal (J-K) ocular NC cells. Squares in G show individual locations of magnified cells in H-K. 1˚NC cells appear flat and rectangular (arrows), while 2˚NC cells show mesenchymal morphology with filopodia-like cellular protrusions (arrowheads). Orientation: dorsal (D), ventral (V), rostral (R) and caudal (C); Ln: lens; Scale bars: 20 μm.
(PDF)

**S7 Fig. Lineage of proximal and distal destined NC cells.** Lineages for each of the proximal (magenta) or distal (green) destined group are shown in tree diagram. Each branch corresponds to a cell division. Incomplete short tracks are shown in blue.
(PDF)

**S8 Fig. 2˚NC cells arise from the dorsal diencephalon and mesencephalon.** Wildtype embryos at 18 hpf were examined for the indicated combinations of diencephalic/mesencephalic marker genes by fluorescence double *in situ* hybridisation. Anterior top, dorsal views. The boundary between diencephalon (d) and mesencephalon (m) is indicated by a stippled horizontal line. (A-B) In dorsal views, both *sox10* and *sox9b* appear co-expressed with *pax6b* that is expressed in the posterior diencephalon. However, co-localization analysis showed neither *sox10*- nor *sox9b*-positive cells co-express *pax6b* (A' and B'). The posterior end of *pax6b* expression marks the boundary between diencephalon and mesencephalon (stippled line). (C, C') *sox9b*-positive cells (magenta) in the diencephalon co-express the epiphysis marker gene, *otx5* (green). Scale Bar: (A-C, A'-C') 50 μm.
(PDF)

**S9 Fig. Mapping of the transgene insertion site of *Tg(-3.1mnx1.1:GFP)$^{ml4}$*.** A genomic DNA library was prepared from 50 embryos and paired-end reads (2×50 nucleotides) were obtained with an Illumina Hiseq1500 sequencer. Two insertion sites, one in chromosome 7 and another in chromosome 9, were identified. By outcrossing into wild type, the insertion site mapped to the end of chromosome 9 on the minus strand (Chr9: 56325279–80, GRCz10) was found responsible for the ectopic corneal endothelium expression of the GFP reporter.
(PDF)

**S10 Fig. Absence or reduced Sox10 activity causes abnormal AS structures.** (A-F) Ultrastructure analysis of the cornea (arrow) from *sox10$^{t3/+}$* or *sox10$^{+/+}$* sibling embryos (A-B, $n = 2$ embryos) and *sox10$^{t3/t3}$* homozygous mutants (C-F, $n = 4$). Low magnification overviews (A, C and E) and respective magnified views (B, D and F) are shown. Stippled line demarcates the lens. *sox10$^{t3/t3}$* homozygous mutants showed vacuolated thick layer of cells (C-D) beneath the corneal stroma (St). Thinning of the same layer with oedema (asterisk) was also observed (E-F). Scale bar: 1 μm (B, D); (G-I) Morpholino knockdown of *sox10* caused abnormal AS with reduced number of the corneal endothelium. (G-H') At 5 dpf the corneal endothelium (grey) forms a monolayer of cells over the lens, which is surrounded by a ring of annular ligament cells (stippled magenta circle) in en face views (G-H). (G'-H') Corresponding transverse section views along the nasotemporal axis of a control and *sox10*-KD embryo shown in G-H. (G'-H') Stippled double-head arc arrows (magenta) show the extent of the corneal endothelium. Note in agreement with the electron micrographs the endothelial cells appear thicker and more irregular than the control endothelial cells in G'. Orientation: nasal (N), temporal (T), dorsal (D) and ventral (V); Scale bar: 50 μm (I) Control embryos at 5 dpf have 28.6 ± 3.6 corneal endothelial cells ($n = 25$), whereas *sox10*-KD embryos show 12.8 ± 6.7 cells ($n = 28$). ***Welch Two sample *t*-test *p*-value = 1.3 x10$^{-13}$ ($t = 10.7$, $df = 42.1$).
(PDF)

**S11 Fig. The *pax6a* mutant locus created by CRISPR/Cas9.** (A) The chromosome 25 *pax6a* locus highlighted with two guide RNA sites in exon 8 and 12 (magenta arrows). Two primer sites for identifying the genomic deletion (double headed arrow) are indicated by the small black horizontal arrows. Large black arrow: direction of transcription. (B) Schematic diagram of a predicted mutant *pax6a* transcript is given in parallel with the wild type transcript. Length of each exon is equalized. Magenta vertical arrows indicate the site of the two guide RNAs. (C) Predicted protein sequence of mutated *pax6a* locus is aligned with wild type *pax6a* sequence. The 3 kb genomic deletion causes a frame shift between paired box and homeobox DNA binding domains. (D) RNAseq analysis of mutant *pax6a* mRNAs. Transcripts in the eye of wild type and *pax6a-/-;pax6b-/-* double mutants at 5 dpf were isolated and processed for library construction for 50-bp paired-end RNA sequencing. Reads mapped across the deletion locus created by exon 8–12 fusion (black boxed region in B) were 1401 RPKM (reads per kilobase of transcript, per million mapped reads) for *pax6a-/-;pax6b-/-* double mutants, whereas only 57 RPKM were mapped here in wild type eyes (24.7-fold increase in *pax6* mutants). In contrast, the number of reads mapped across the exon 12–13 boundary (white boxed region in B), which were expected to be equal between embryos from both genotypes, was comparable between the two groups (1.18-fold decrease in *pax6* mutants). (E-F) *in situ* analysis of gene expression of *pax6a* (E-E'") and *pax6b* (F-F'") in the eye at 28 hpf with wild type (WT) embryos (E-F), zygotic (Z) *pax6b-/-* mutants (E'-F'), maternal-zygotic (MZ) *pax6b-/-* mutants (E"-F") and *pax6a-/-;pax6b-/-* double mutants (E'"-F'"). Transverse sections at the level of the lens (or the equivalent cavity denoted by an asterisk for the lensless *pax6a-/-;pax6b-/-* mutants) are shown. Dorsal side is up with the distal side oriented to the left. The boundary of the eye and the lens are indicated by the stippled lines. In the WT, the lens epithelium located on the distal end expresses *pax6a* (E, black arrow) and *pax6b* (F, black arrow). In zygotic *pax6b-/-* embryos, *pax6a/b* genes are mis-expressed in the proximal side of the lens (E' and F', white arrows). The maternal-zygotic *pax6b-/-* embryo that shows the small lens phenotype with higher penetration than the zygotic mutant exhibit the same abnormal polarization of *pax6a* (E", white arrow) and *pax6b* (F", white arrow) in the lens. In *pax6a-/-;pax6b-/-* mutants the eye shows normal levels and distribution of *pax6a* (E'") and *pax6b* (F'") albeit the absence of the lens primordium (asterisk). Thickness of the section: 6 μm; Scale bar: 100 μm. (PDF)

**S12 Fig. *pax6ab* antisense morpholino oligonucleotide injection mimics the genetic mutant phenotype.** (A) Protein sequence alignment of human PAX6 and zebrafish Pax6 proteins. A part of the N-terminal region (first 87 amino acids from human PAX6) is shown. Note that zebrafish Pax6 proteins have two in-frame methionine residues (marked with orange) and the second translation start site is conserved in the human genes. Location of exon2 (Ex2) and exon3 (Ex3) for zebrafish Pax6 proteins are indicated below the sequence. (B) CAGE tag distribution for zebrafish *pax6b* locus [77]. Results from two stages (somitogenesis stages and 33 hpf) relevant to the anterior chamber formation are shown. Note that the two regions (magenta rectangle) corresponding to the first and second methionine of the Pax6b protein are used as transcription start sites, indicating the necessity to block both transcripts for effective knockdown of Pax6b protein. Two antisense morpholino oligonucleotides (MOs) used in this study (1ATG_pax6ab: 5'-GTTATGGTATTCTTTTTGAGGCATT-3'; 2ATG_pax6ab: 5'-ACTGTGACTGTTTTGCATCATGGAC-3', see also Supplemental Methods) target both Pax6b transcripts as well as Pax6a. (C-D) Knockdown efficiency of *pax6a/b* MOs was tested on a reporter construct that harbours two MO target sites (1st and 2nd ATG region) upstream of the GFP coding sequence. mRNA of the reporter construct was synthesized by SP6 RNA

polymerase and injected into one-cell stage wildtype zebrafish embryos with or without *pax6a/b* morpholinos (MOs). Injected embryos were raised until 10 hpf and GFP fluorescence was examined under a stereomicroscope (C) and quantified with the image analysis software ImageJ (D). *pax6a/b* MOs reduced GFP intensity significantly in comparison to injection of the reporter construct alone ($p<0.001$, D). Scale bar: 1 mm. (E-F) The Sox10 reporter mEos was converted from green to red (magenta) at 20 hpf in embryos injected with either mismatch control (E) or *pax6a/b* (F) MOs. This protocol led to selective labelling of 1˚ and 2˚NC cells in magenta and green, respectively. Embryos were examined at 27 hpf. 2˚NC cells were detected as cells with green fluorescence in the distal eye compartment in control embryos (arrowhead, E) while 1˚NC cells remained in magenta at the proximal side of the eye (asterisk, E). The distal compartment of *pax6a/b* MO injected embryos (morphants) is, however, devoid of mem-tdEosFP expressingcells (F), although 1˚NC cells were found unaffected at the proximal side of the eye (asterisk, F). (G-H) Embryos injected with control MOs show NC cells with endogenous *sox10* mRNA expression (arrowhead) in the distal compartment of the eye (white circle, $n = 14/14$ embryos). In contrast, *sox10* mRNA expressing NC cells were not observed in the distal compartment (white circle) after *pax6a/b* knockdown ($n = 15/15$). Scale bar: 50 μm. (PDF)

**S13 Fig. Nasotemporal polarity of the eye is not affected in pax6a/b double mutants.** Embryos (28 hpf) were hybridized to *foxd1* and *foxg1a* DIG-labelled antisense RNA probes. Arrows point at normal expression of nasal (A-A") and temporal (B-B") expression of *foxg1a* and *foxd1*, respectively. Dorsal views anterior left. Scale bar: 50 μm. (PDF)

**S14 Fig. Nrp2b is required for correct guidance of NC cells and corneal endothelium formation.** (A-B) NC cells (green) that migrated into the distal side of the eye form a NC-free area over the lens delineated by leading edge cells forming an open ring structure (stippled white line). Note that NC cells are absent in the ventral aspects of the distal eye at 40 hpf in the embryos injected with a control morpholino (A, $n = 14$ embryos). Upon *nrp2b* morpholino knock-down (KD), the NC-free area over the lens became smaller (B, stippled white circle; $n = 17$ embryos) with ectopic NC cells in the ventral part of the distal eye (arrow). Orientation: nasal (N), temporal (T), dorsal (D) and ventral (V). (C-D') At 5 dpf the corneal endothelium (grey) forms a monolayer of cells over the lens, which appear surrounded by a ring of annular ligament cells (stippled magenta circle) in en face views (C-D). (C'-D') Corresponding transverse section views along the nasal-temporal axis of a control and *nrp2b*-KD embryo shown in C-D. Stippled double-head arc arrows (magenta) show the extent of the corneal endothelium. Scale bar: 50 μm (E) Corneal endothelial cells of control embryos at 5 dpf are composed of $25.3 \pm 3.8$ cells ($n = 22$), whereas those of *nrp2b*-KD embryos are $4.8 \pm 2.7$ cells ($n = 28$). ***Welch Two sample *t*-test *p*-value$<2.2 \times 10^{-16}$ ($t = 20.9$, $df = 36.3$). (PDF)

**S1 Method. Morpholinos and transmission electron microscopy.** (DOCX)

**S2 Method. All numerical data underlying graphs and statistics.** (TXT)

## Acknowledgments

We thank N. Borel and our fish house team, M. Rastegar for microscopy, Drs. R. Kelsh, F. Argenton, B. Peers and L. Lum for embryos and reagents.

## Author Contributions

**Conceptualization:** Masanari Takamiya, Uwe Strähle.

**Formal analysis:** Johannes Stegmaier, Benjamin Schott, Victor Gourain, Tim Scherr.

**Funding acquisition:** Uwe Strähle.

**Investigation:** Masanari Takamiya, Andrei Yu Kobitski, Benjamin D. Weger, Dimitra Margariti, Angel R. Cereceda Delgado, Lixin Yang, Sebastian Sorge, Jens C. Otte.

**Project administration:** Sepand Rastegar, Ralf Mikut, Gerd Ulrich Nienhaus.

**Resources:** Volker Hartmann, Jos van Wezel, Rainer Stotzka.

**Writing – original draft:** Masanari Takamiya, Uwe Strähle.

**Writing – review & editing:** Masanari Takamiya, Thomas Reinhard, Günther Schlunck, Thomas Dickmeis, Ralf Mikut, Gerd Ulrich Nienhaus, Uwe Strähle.

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
