## [Decision Letter · Decision Letter 0]

8 Oct 2019

Dear Dr Strähle,

Thank you very much for submitting your Research Article entitled 'Pax6 organizes the anterior eye segment independently of optic cup patterning by guiding two distinct neural crest waves' to PLOS Genetics. Your manuscript was fully evaluated at the editorial level and by independent peer reviewers. The reviewers appreciated the attention to an important problem, but raised some substantial concerns about the current manuscript. Based on the reviews, we will not be able to accept this version of the manuscript, but we would be willing to review again a much-revised version. We cannot, of course, promise publication at that time.

If you decide to revise the manuscript for further consideration at PLOS Genetics, please aim to resubmit within the next 60 days, unless it will take extra time to address the concerns of the reviewers, in which case we would appreciate an expected resubmission date by email to plosgenetics@plos.org.

[LINK]

We are sorry that we cannot be more positive about your manuscript at this stage. Please do not hesitate to contact us if you have any concerns or questions.

Yours sincerely,

Brian A Link

Guest Editor

PLOS Genetics

Gregory Barsh

Editor-in-Chief

PLOS Genetics

Reviewer's Responses to Questions

**Comments to the Authors:**

Reviewer #1: In this manuscript, Takamiya et al. aim to address a number of fundamental questions related to cranial neural crest and contributions to eye development. The authors use timelapse imaging and cell tracking to conclude that there are two waves of NC migration around the eye. They find that zebrafish pax6 mutants exhibit defects in anterior segment development, and NC cells appear to migrate aberrantly. pax6 mutant eyes exhibit disrupted expression of molecules known to affect cell migration, as well as defects in the lens, which is known to play an organizing role in the anterior segment.

There is a substantial amount of data in this manuscript, addressing an important and understudied aspect of eye development, however, in this form, there are concerns regarding some of the authors’ conclusions, related to a number of different key points in the manuscript. General concerns and specific points are detailed below.

General comments:

• The language is confusing with respect to descriptions of the different NC populations, cell tracking and cell migration. For example, the word “origin” is used to describe a specific position for the secondary NC, yet the cell tracks shown go further back in time, and to a similar spot as the primary NC. If the authors consider this an “origin” with respect to sox10 transgene expression, the language should be made more precise. With respect to cell migration, migration “modes” and transformations are referred to, especially in the discussion (starting at line 385), and such data are not adequately presented in the manuscript to generate such conclusions; single cell behaviors are not resolvable.

• A direct correlation between sox10 transgene-positive cells and cornealEndo transgene-positive cells is missing: it is unclear how many of the AS cells are actually primary vs secondary. If the authors want to draw conclusions in which NC waves are connected to the corneal endothelium, this is an important point. I understand that there is a technical issue in which the authors may not have the appropriate tool in hand to do bona fide lineage tracing; in this case, caution should be exercised with respect to the conclusions that can be drawn.

• In the pax6 mutants, the authors examine the expression of some signaling pathway components known to affect cell motility and guidance. The authors conclude that a function of pax6 is in guidance of neural crest migration, yet no direct test is carried out. If the authors are hypothesizing that changes in the expression of these guidance molecules are affecting neural crest migration, the authors could manipulate some of these genes (via mutants or morpholinos) to directly test a role in guiding primary or secondary neural crest migration.

• In general, the manuscript is a bit unfocused and difficult to read. For example, how exactly does wnt signaling actually fit into this? The experiments presented suggest that it does not play a role in AS development. Another example is Figure 7, in which the TGF-beta signaling pathway is examined, yet this is unconnected to the rest of the manuscript. Can TGF-beta be connected to the rest of the manuscript by testing it in the context of pax6 mutants, or adding it back to rescue pax6 mutants?

Specific points:

• Figures 1 and 2: the authors use high resolution imaging and cell tracking to examine neural crest migration around the eye. They conclude that there are two waves of NC migration, with different origins and destinations. The difference in “origins” is confusing, as the tracks in Figure 2 appear to extend back to a start position similar to the primary cells. The difference in destinations is clear at 31 hpf, yet both populations may contribute to the corneal endothelium shown in Figure 1J-L’. Can the authors comment on their confidence in the distinctions between these origins and endpoints? Is there a lineage tracing tool that can be used (sox10:Cre combined with a zebrabow type reporter) to identify the endpoints and contributions more definitively?

• Figure 2 and S3: the forward tracking of the primary NC (2D-D’ and S3D-D’) appear somewhat different from each other in the two different embryos. In the second embryo, are the cells all converging toward a specific spot behind the eye?

• Figure 1 and 2 and conclusions summarized in the Discussion (lines 385-395): the authors use their imaging to describe some of the migration behaviors of the primary and secondary NC populations. These behaviors are difficult to see, since single cell morphologies are not really resolvable; unmerged Eos images would be helpful to include to assess migration patterns and behaviors. The Discussion contains some broad statements about transformations of collective cell behavior, “mesenchymal” cell shapes, and use of the word “misrouting”; the data presented in this manuscript are not adequate to support these conclusions. It would be helpful to quantify these phenotypes at the single cell level.

• Figure 1J-L: the authors use the corneaEndo:GFP transgenic line. More validation is necessary to demonstrate that this is corneal endothelium: in Figures 1L’-L’’, a small arrow is placed to mark the epithelium, but what this arrow points to is difficult to see. In addition, it is not clear exactly how the cells marked by this transgene are related to the sox10 transgene positive cells.

• Figure 1N-O: the sox10 mutant cornea phenotype is difficult to interpret. The authors state that the endothelial layer is missing. Can this be quantified? For example, is the number of corneal endothelial cells different? Or the thickness of the corneal endothelium? Can the authors zoom out to show the phenotype in a broader view?

• Figure S6: the implications of the wnt inhibitor experiments are unclear. Treatment with IWR-1 does affect secondary neural crest cells, but not formation of the corneal endothelium. Does this mean that the secondary neural crest cells are not important for the corneal endothelium?

• Figure 4A-B: the authors use tdEos photoconversion to examine when the sox10 transgene is turned off. Strictly speaking, this experiment examines when translation of tdEos stops, not transcription; the title of the chart in 4B should be edited.

• Table 1 and Figure 6: the authors argue that Pax6a/b dose correlates with NC migration and anterior segment defects. Neither dosage nor phenotypes were quantified beyond numbers of wild type or mutant alleles, or in the case of phenotypes, numbers of embryos. Given recent studies of transcriptional compensation in mutant backgrounds, if dosage is an crucial point, it would be important to know quantitatively how pax6 expression and function might be affected in each genotype (for example, using qRT-PCR to determine level of expression, or luciferase assays to determine function of the missense mutant). In terms of phenotypes, was the number of cells exhibiting migration defects affected? Or the just the numbers of embryos?

• Figure 4: the authors use in situ hybridization of a number of different eye genes to conclude that patterning is not affected in the pax6 mutants. The data (specifically the genes assayed) suggest that dorsoventral and proximodistal patterning are not affected, but the statement that patterning is “unaffected” is a bit of an overreach in the absence of nasotemporal markers (such as foxg1 and foxd1).

• Figure 4: the authors find alterations to cell guidance molecules in the pax6 mutants, and a major conclusion is that pax6 guides the neural crest. The data do not necessarily support a role for guidance: If the authors hypothesize that some of these molecules are important for NC guidance, expression of these molecules should be directly manipulated in a wild type background (via genetic mutation or morpholino oligonucleotide) and NC migration examined.

• Figure 7: the TGF-beta-related experiments in this figure seem unrelated to the rest of the manuscript. Because the lens transplant seemed to rescue some of the AS phenotypes in Figure 6, can the authors use a TGF-beta soaked bead to see if TGF-beta is sufficient to rescue the phenotype? Or examine sox10 and pitx2 expression in embryos treated with the TGF-beta inhibitor?

• Discussion: there are statements that appear to be slightly over-reaching, based on the data presented in the manuscript. These include language regarding cell behavior and “transformations” thereof (lines 385-395); this is not directly shown or addressed in this manuscript, as the transgenic lines do not provide enough cellular resolution to make these conclusions. The word “misrouted” (line 395) is strong and suggests guidance to a different location, which is unclear.

Minor points:

• Figure 3: the term DMB should be defined

• Line 325-326: “Lens-derived TGF-beta2 organizes periocular mesenchyme differentiation to form the AS in mice.” This statement needs a citation.

Reviewer #2: This manuscript [PGENETICS-D-19-01379] by Takamiya et al. details a major research advance in our understanding of neural crest cells in the vicinity of the developing eye. The researchers demonstrate that there are indeed two separable populations of neural crest cells in this region. Furthermore, they demonstrate the key roles of Pax6 and Tgf-b in this process. Overall, this is a fascinating story and has widespread implications for the analysis of anterior segment development and the studies of anterior segment dysgenesis. Although the vast majority of experiments are well controlled and convincing, there are a few outstanding issues that remain (all outlined below):

Major:

1. The examination of patterning and guidance molecule gene expression seems rather confusing. It is quite obvious that expression of guidance molecule genes is strongly affected. The conclusion, though, that patterning is unaffected seems unsupported by the data. Tbx5, aldh1a2, and vax2 show obvious, albeit mild, changes to gene expression. Yet the authors conclude that patterning is unaffected by pax6. I suggest that a quantitative approach would be required to show that such patterning genes are unaffected (qPCR of retinal tissue and/or measurement of area of gene expression). If indeed such genes are affected, the authors will need to rework their conclusions and their title.

2. A functional correlation between guidance molecule function and neural crest migration has not been established. To establish this, the authors are encouraged to perform either rescue experiments (Tg[eye promoter::guidance gene] to rescue phenotype of mutants) or knockdown (or pharmacological block) to inactivate cues.

3. The analysis of tgfb function seems somewhat incomplete. The authors have not analyzed a neural crest reporter in such experiments and this is imperative. The authors should also determine the critical period of Tgf-b function by performing treatments at differing stages.

Minor:

1. The eye has changed size in Fig 7, which prevents the reader from seeing periocular gene expression in the later stages. The authors should increase the size of the pictures such that the entire eye is shown.

2. The genetic background has recently been shown to play a considerable role in phenotypic penetrance. Although it is possible that this was missed in the text, I didn’t see a mention of the genetic background for the experiments performed (AB, TB, or other). This should be included. Also, did the researchers perform any experiments on other backgrounds?

**Have all data underlying the figures and results presented in the manuscript been provided?**

Reviewer #1: Yes

Reviewer #2: Yes

PLOS authors have the option to publish the peer review history of their article (what does this mean?). If published, this will include your full peer review and any attached files.

Reviewer #1: No

Reviewer #2: Yes: Andrew Jan Waskiewicz

---

## [Decision Letter · Decision Letter 1]

17 Feb 2020

Dear Dr Strähle,

Thank you very much for submitting your Research Article entitled 'Pax6 organizes the anterior eye segment by guiding two distinct neural crest waves' to PLOS Genetics. Your manuscript was fully evaluated at the editorial level and by independent peer reviewers. The reviewers appreciated the attention to an important topic but identified some aspects of the manuscript that should be improved.

We therefore ask you to modify the manuscript according to the review recommendations before we can consider your manuscript for acceptance. Your revisions should address the specific points made by each reviewer.

[LINK]

Yours sincerely,

Brian A Link

Guest Editor

PLOS Genetics

Gregory Barsh

Editor-in-Chief

PLOS Genetics

Reviewer's Responses to Questions

**Comments to the Authors:**

Reviewer #1: The authors have made a significant effort to answer the questions and comments posed. The reorganization of the data, and the edits to the figures are helpful, as is the thorough point-by-point response.

I have a few further questions and comments:

• Regarding the new TGFbeta2 data included in the reviewer response: I think the partial rescue is actually quite exciting, and could be included as supplementary data. I don’t think that anyone would argue that TGFbeta2 is the only relevant factor coming from the lens; the data demonstrate that TGFbeta2 is a piece of the puzzle downstream of pax6a/b and the lens.

• Inclusion of cell movement data is helpful (S6 Figure), however, this still doesn’t really answer the question of cell morphology. In the previous review (points 10 and 11), I had concerns regarding the use of terms related to descriptions of cell behavior (e.g. “collective behavior”, “mesenchymal cell shapes”). S6 Figure appropriately addresses directionality and processivity of movement. Extra movies are helpful (e.g. S9 Video), but without more single cell morphology information, collective cell behavior is still somewhat hard to see.

• p. 9, line 217-218, “Corneal endothelial cells proliferate until they occupy the entire inner surface of the cornea.” Did the authors show that proliferation is the driving mechanism?

Reviewer #2: The authors have addressed all of my concern. Notably they are to be applauded for their extremely thorough rebuttal letter and many additional experiments included. Only one textual issue was apparent. In the morpholine experiments (S14), did they use published MO sequences. Perhaps this could be clarified in methods or in results section.

**Have all data underlying the figures and results presented in the manuscript been provided?**

Reviewer #1: Yes

Reviewer #2: Yes

PLOS authors have the option to publish the peer review history of their article (what does this mean?). If published, this will include your full peer review and any attached files.

Reviewer #1: No

Reviewer #2: Yes: Andrew J. Waskiewicz

---

## [Decision Letter · Decision Letter 2]

9 Apr 2020

Dear Uwe,

We are pleased to inform you that your manuscript entitled "Pax6 organizes the anterior eye segment by guiding two distinct neural crest waves" has been editorially accepted for publication in PLOS Genetics. Congratulations!

Yours sincerely,

Brian A Link

Guest Editor

PLOS Genetics

Gregory Barsh

Editor-in-Chief

PLOS Genetics

Comments from the reviewers (if applicable):

Reviewer's Responses to Questions

**Comments to the Authors:**

Reviewer #1: I appreciate the authors’ efforts and responses. I have no further questions now.

**Have all data underlying the figures and results presented in the manuscript been provided?**

Reviewer #1: Yes

PLOS authors have the option to publish the peer review history of their article (what does this mean?). If published, this will include your full peer review and any attached files.

Reviewer #1: No

**Data Deposition**

http://datadryad.org/submit?journalID=pgenetics&manu=PGENETICS-D-19-01379R2

**Press Queries**

---

## [Editor Report · Acceptance letter]

8 May 2020

PGENETICS-D-19-01379R2 

Pax6 organizes the anterior eye segment by guiding two distinct neural crest waves 

Dear Dr Strähle, 

We are pleased to inform you that your manuscript entitled "Pax6 organizes the anterior eye segment by guiding two distinct neural crest waves" has been formally accepted for publication in PLOS Genetics! Your manuscript is now with our production department and you will be notified of the publication date in due course.

With kind regards,

Kaitlin Butler

PLOS Genetics

On behalf of:
